# Ontology-Retrieval Augmented Generation for Scientific Discovery

## Abstract

Large Language Models (LLMs) have demonstrated remarkable capabilities across a wide range of tasks, sparkling an increasing interest for their application in science. However, in scientific domains, their utility is often limited by hallucinations that violate established relationships between concepts or ignore their meaning; problems that are not entirely eliminated with Retrieval Augmented Generation (RAG) techniques. A key feature of science is the use of niche concepts, abbreviations and implicit relationships, which may deem RAG approaches less powerful due to the lack of understanding of such concepts, especially in emerging and less known fields. Ontologies, as structured frameworks for organizing knowledge and establishing relationships between concepts, offer a potential solution to this challenge. In this work we introduce OntoRAG, a novel approach that enhances RAG by retrieving taxonomical knowledge from ontologies. We evaluate the performance of this method on three common biomedical benchmarks. To extend the value of OntoRAG to emerging fields, where ontologies have not yet been developed, we also present OntoGen, a methodology for generating ontologies from a set of documents. We apply the combined OntoGen+OntoRAG pipeline to a novel benchmark of scientific discovery in the emerging field of single-atom catalysis. Our results demonstrate the promise of this method for improving reasoning and suppressing hallucinations in LLMs, potentially accelerating scientific discovery across various domains. The code for reproducing the results is available under https://figshare.com/s/4f898ef092ae5898c1b7.

## 1 Introduction

Scientific discovery stands at the forefront of human progress, driving innovations across fields from medicine to materials science (Daston, 2017). In recent years, Large Language Models (LLMs) have emerged as promising tools to accelerate this process, offering unprecedented capabilities in natural language understanding and generation (Vaswani et al., 2017). In view of these capabilities, a growing volume of work has been devoted towards applying LLMs and derived technologies into a number of scientific domains (Birhane et al., 2023). Despite these efforts, the application of LLMs to complex scientific domains remains challenging, often hampered by the models' tendency to generate plausible but factually incorrect information, a phenomenon collectively referred to as "hallucinations" (Verspoor, 2024).

Retrieval Augmented Generation (RAG) has been proposed as a solution to ground LLM outputs in factual information by providing relevant documents during inference (Lewis et al., 2020), leveraging the in-context learning (ICL) capabilities of these models (Olsson et al., 2022). While RAG has shown promise in general knowledge domains (Shuster et al., 2021; Lewis et al., 2020), its application to specialized scientific fields often falls short (Soong et al., 2024). The complexity and interconnectedness of scientific concepts (Bizon et al., 2019), coupled with the rapid evolution of knowledge in cutting-edge fields (Agrawal & Choudhary, 2016), pose significant challenges to traditional RAG approaches (Wu et al., 2024).

In this paper, we introduce OntoRAG, a novel methodology that combines the power of ontologies with RAG to enhance applications in scientific discovery. Ontologies, as formal representations of domain knowledge, offer a structured framework of concepts and relationships (Keet, 2018) that can guide LLM reasoning. By integrating ontological knowledge into the retrieval and generation process,

Figure 1: Overview of the OntoRAG methodology and evaluation framework. The key components of OntoRAG include document retrieval, and the integration of ontologies for enhanced reasoning in specific fields. OntoGen automates the process of building ontologies from research document corpora. The system is evaluated through SACBench, a benchmark for evaluation of synthesis planning of Single-Atom Catalysts, demonstrating its application in cutting-edge scientific domains.

OntoRAG aims to produce more accurate, logically consistent, and scientifically grounded outputs. This approach is particularly advantageous in emerging scientific fields where the terminology, definitions, and conceptual relationships are often still evolving and may not be well-represented in the training data of general-purpose LLMs. This capability addresses a significant limitation of standard RAG approaches, which may underperform when dealing with cutting-edge scientific concepts and terminology.

A significant bottleneck in ontology-based systems is the time-intensive and expert-dependent process of ontology creation (Keet, 2018). Tools have been proposed recently to accelerate this process by inserting new terms into an already existing ontology Toro et al. (2023); Funk et al. (2023), or automating tasks such as term typing, taxonomy discovery, etc, under the frame of Ontology Learning Ciatto et al. (2024); Toro et al. (2023); Babaei Giglou et al. (2023). However, none of these works have attempted to generate full ontologies. To address this we propose OntoGen, an LLM-based method for automatic end-to-end generation of ontologies. Our pipeline iteratively constructs domain-specific ontologies from collections of relevant scientific documents, making the OntoRAG approach adaptable to new and rapidly evolving fields of study. This enables the application of ontology-based methods to a wider range of scientific domains, and in particular fast-developing fields and those where no ontologies have yet been defined. The full pipeline (Figure 1) relies entirely on a predefined corpus of papers that determines the scope of the field and produces both a RAG application and an ontology, which may have many other use-cases (Jablonka et al., 2024).

To demonstrate the efficacy of OntoRAG, we apply the pipeline to a novel benchmark in the emerging field of Single-Atom Catalysis (SAC). This cutting-edge area of materials science serves as an ideal case study due to its novelty, complexity, and the critical role of synthesis procedures in advancing the field. While SAC holds promise for more efficient and selective catalytic processes Wang et al. (2018), the challenge of predicting effective synthesis methods often bottlenecks progress Grzybowski et al. (2023). Additionally, we perform control evaluations on three standard biomedical benchmarks for Q&A (MedMCQA Pal et al. (2022), MedQA Zhang et al. (2018), and MMLU-Medical (Hendrycks et al., 2020; Chen et al., 2023b)), where we observe no decrease in accuracy relative to baselines.

Our contributions are threefold:

- We present a novel ontology generation pipeline that automatically constructs domain-specific ontologies from scientific literature (OntoGen).

- We introduce OntoRAG, a methodology for ontology-based Retrieval Augmented Generation that enhances the scientific accuracy of LLM outputs.

- We develop a literature-based benchmark for the generation of synthesis procedures in the field of Single-Atom Catalysis, providing a concrete evaluation framework for our approach.

The results indicate not only the potential that LLMs offer for ontology engineering, in the sense of rapid prototyping and in niche domains, but also the usefulness that ontologies provide to LLM-based applications as a source of structured and human interpretable information about said domains. Our experiments show that OntoRAG outperforms traditional RAG methods in generating accurate and logically consistent scientific content, especially on fast-growing fields. Our results suggest that ontology-based approaches can play a crucial role in adapting LLMs for specialized scientific tasks, potentially accelerating the pace of discovery across various scientific domains.

## 2 BACKGROUND

### 2.1 ONTOLOGIES

From a practical engineering perspective, ontologies are a specification of conceptualization using description logic (Iqbal et al., 2013). They model relationships between concepts and the words used to represent them. In their most basic form, an ontology can be thought of as a series of "isA" relationships formed in a hierarchical tree structure called a taxonomy (Keet, 2018). This taxonomy serves as the backbone of an ontology and can be engineered to include other relationships, instances, properties, and axioms. For further reading on the definition of an ontology, the reader should reference "What is an Ontology" Guarino et al. (2009) and "An Introduction to Ontology Engineering" Keet (2018), however for the purpose of this work, we use the definition from Guarino et al. (2009): "an ontology is an explicit conceptualization of knowledge." When engineering ontologies for the sake of our workflow, we utilize the following definition.

**Definition 2.1** *An ontology is a tuple $\{\mathcal{C}, \mathcal{R}, \mathcal{I}, \mathcal{P}\}$ where:*

- *$\mathcal{C}$ is a set of classes $\{C_1, C_2, ..., C_n\}$ present in the ontology.*

- *$\mathcal{R}$ is a set of relationships present in the ontology.*
  *$\mathcal{R} = \{(C_i, r_s, C_j) | r_s \in \mathcal{R}_s\}$, where $\mathcal{R}_s$ is the set of all possible relations (object properties).*

- *$\mathcal{I}$ is the set of all instances of classes present in the ontology*
  *$\mathcal{I} = \{i_1, i_2, ..., i_m\}; i : C, C \in \mathcal{C}$.*

- *$\mathcal{P}$ is the set of all possible properties in an ontology.*
  *$\mathcal{P} = \{p_1, p_2, ..., p_l\}$ and $p : \mathcal{I} \to \mathcal{V}$ or $p : \mathcal{C} \to \mathcal{V}$; where $\mathcal{V}$ is the set of all possible values for a property (data properties).*

The creation of ontologies is a complex process that typically involves domain experts, knowledge engineers, and extensive evaluations (Iqbal et al., 2013). The development process often relies on domain analysis and knowledge acquisition, identification of key concepts and relationships, followed by formal representation of concepts, relationships, and axioms (Keet, 2018; Contreras et al., 2019). Ontologies then require extensive validation and refinement, as well as documentation and maintenance (Yang et al., 2019).

In scientific domains, ontologies play a crucial role in organizing and standardizing knowledge. Some notable examples include the Gene Ontology (GO) (Aleksander et al., 2023), Chemical Entities of Biological Interest (ChEBI) (Degtyarenko et al., 2007), and NASA's Semantic Web for Earth and Environmental Terminology (SWEET) (Raskin & Pan, 2005). Beyond organizing knowledge, ontologies have found numerous practical applications including data integration (Confalonieri & Guizzardi, 2024), information retrieval (Vallet et al., 2005), natural language processing (Abburu & Golla, 2017), building decision support systems (Bhattacharyya, 2015), among others (Jain & Kihara, 2019).

### 2.2 RETRIEVAL AUGMENTED GENERATION

Retrieval Augmented Generation (RAG) is a hybrid approach that combines the strengths of retrieval-based and generative models in natural language processing (Lewis et al., 2021). It enhances

the capabilities of LLMs by providing them with relevant external knowledge to supplement the information provided in an input prompt. RAG was originally defined in Lewis et al. (2021) . Since the original work, several other methods of RAG have been evoked in the literatureWang et al. (2024) Shahul et al. (2023). In this work, we define RAG in terms of a retrieval function and a fusion operator, defined by:

$$p(y|x) = p_\theta(y|F(x, R(x))), \qquad \text{with} \qquad R(x) = \arg \max_{z \in Z}^{k} \{r(z, x)\} \qquad (1)$$

Where:

- $p(y|x)$ is the probability of generating output y given input x.
- $R(x)$ hence defines a set of the $k$ most relevant documents to $x$ under relevance function $r$.
- $r$ is a *document relevance* function, such that $r(z, x)$ quantifies the relevance of document $z$ to query $x$.
- $F$ is a fusion operator.
- $p_\theta(y|w)$ is the probability of generating $y$ given context $w$ for a language model parameterized by $\theta$.

Thereby conditioning the output of an LLM on both the input $x$ (e.g. a question) and a set of documents $z$ with retrieval relevance $r(z|x)$.

The role of the fusion operator $\mathcal{F}$ is to combine retrieved documents $z$, along with input $x$, in such a way that it can be given as a prior to the model, conditioning generation (Izacard & Grave, 2021). Such operator can take several forms, and its influence on performance has been documented elsewhere Liu et al. (2024a). This operator can also in principle be optimized thanks to SOTA frameworks Khattab et al. (2023); Yin (2024); Hou et al. (2022).

The document relevance function $r(z|x)$ can take many forms, encompassing both dense and sparse retrieval methods. Dense retrieval methods (Lewis et al., 2020; Xiong et al., 2020) rely on neural encoder models to map both queries and documents into a shared embedding space, enabling semantic matching. These approaches have seen significant advancements in domain-specific applications, such as RXNFP (Schwaller et al., 2021) for chemical reactions and MatBERT (Trewartha et al., 2022) for materials science. On the other hand, sparse retrieval methods like BM25 (Robertson et al., 2009) and TF-IDF (Salton & Buckley, 1988) are based on statistical features of the data (Robertson et al., 2009). These approaches often excel in scenarios requiring exact matching or when dealing with out-of-distribution queries (Luan et al., 2021).

RAG has found applications in various domains, including question answering (Q&A) (Izacard & Grave, 2021), content generation (Edge et al., 2024), and code generation (Bhattarai et al., 2024) among others (Wang et al., 2024). In science, RAG techniques have been explored for scientific Q&A (Lála et al., 2023), explainable AI (Shahul et al., 2023), and literature review assistance (Edge et al., 2024). Challenges remain in applying RAG to scientific domains, including ensuring the reliability of retrieved information and handling domain-specific terminology (Barnett et al., 2024). Moreover, the evaluation of RAG systems, particularly in specialized scientific contexts, remains a significant challenge, necessitating the development of robust and domain-specific evaluation methodologies.

## 2.3 EVALUATION OF RAG SYSTEMS

The evaluation of Retrieval Augmented Generation (RAG) systems presents unique challenges, particularly in scientific domains where accuracy and reliability are paramount (Soong et al., 2024). Traditional evaluation metrics for natural language processing tasks often fall short in capturing the nuanced performance of RAG systems, especially when domain-specific knowledge is involved (Barnett et al., 2024).

Evaluating RAG methods typically involves assessing both the retrieval and generation components Shahul et al. (2023). Metrics such as Precision, Recall, and Mean Average Precision (MAP) are commonly used to assess the quality of retrieved documents (Robertson et al., 2009). Automated

metrics like BLEU (Papineni et al., 2002), ROUGE (Lin, 2004), and METEOR (Banerjee & Lavie, 2005) are often employed to evaluate the quality of the generated text, however, these often fall short to assess the semantic quality of a text given a question. The more recent BERTScore (Zhang et al., 2019) aims to capture semantic similarity beyond surface-level matches.

**Evaluation of Scientific RAG**  In scientific contexts, additional complexities appear, as now answers need not only to be semantically similar or use similar concepts as a ground truth answer, but they should also make use of those concepts in a logical and scientifically accurate manner. Accurate evaluation thus often requires domain experts, making large-scale automated evaluation challenging (Chang et al., 2024). Assessing the factual correctness of generated content is crucial in scientific applications but is difficult to automate (Petroni et al., 2020). Furthermore, from the perspective of scientific discovery, evaluating the novelty and impact of new hypotheses is hard to quantify (Hope et al., 2020).

Some benchmarks exist that aim to tackle these challenges. KILT (Petroni et al., 2020) is a comprehensive benchmark for knowledge-intensive language tasks, including some scientific domains, SciFact is a dataset for evaluating scientific claim verification, which can be used to assess the factual accuracy of RAG systems in scientific contexts (Wadden et al., 2022). Other works default to expert-in-the-loop evaluation by combining automated metrics with expert assessment (Chang et al., 2024).

Despite these advancements, the evaluation of RAG systems in scientific domains remains an active area of research (Barnett et al., 2024). Challenges persist in developing evaluation methodologies that can accurately assess the reliability, relevance, and potential impact of RAG-generated content across diverse scientific fields (Li, 2024).

## 3 ONTORAG

Despite the success of RAG in mitigating hallucinations of LLMs (Tonmoy et al., 2024), issues still persist. From 1, it is inferred that the approach necessitates of a generative model $p_\theta$ that can adequately adapt to the new context and use the new information correctly. This is not necessarily guaranteed in scientific fields, especially for emerging ones, as models may not be familiar with the terminology, concepts, and relations between them. Additionally, LLMs may struggle to create coherence between different pieces of retrieved information, especially when dealing with complex or nuanced scientific concepts Zhao et al. (2024). This is particularly problematic in domains like chemistry, where precise and accurate information is crucial.

Here we introduce OntoRAG, a variation of RAG that includes information from ontologies to supplement traditional RAG pipelines with taxonomical as well as definitional knowledge. OntoRAG builds on the definition of RAG (eq. 1) by including two new objects: an ontology retriever operator $r_\mathcal{O}$ for ontology $\mathcal{O}$ and a new fusion operator $F$ whose roles are, respectively, to retrieve relevant information from the ontology, and to combine it with the context provided to the generative model. Equation 1 is thus updated as:

$$p(y|x) = p_\theta(y|F(x, R(x), R_O(x))),\qquad(2)$$

with

$$R_O(x) = \{O(c) : c \in C(x)\}\qquad(3)$$

Where Eq. 1 is modified in Eq. 2 to include:

- $R_O(x)$ is the ontological context relevant to query $x$, which depends on:
- $O(c)$ is some ontological context retriever, and
- $C(x)$ is a set of concepts found in text $x$.

In our implementation, querying $r_{\mathcal{O}}$ triggers a Named Entity Recognition (NER) pipeline to recognize concepts in the input $x$, that are found in ontology $\mathcal{O}$, see Appendix A.3. These concepts are then enriched with their parents, children, and definitions in the context of the ontology, which provides relevant context about the role of this concept and its relations with other concepts.

This integration of ontologies into the RAG pipeline provides a finite vocabulary that surrounds a subject domain (such as medical terminology or catalysis), and a structure denoting how these terms relate to one another. Although this vocabulary and relationships *could* potentially be learned by an LLM during training, training data often lacks to effectively denote them on the same level of experts, especially for niche and emerging fields for which there is little publicly available training data. In fact, these information sources can be combined flexibly to leverage the ontological information in several stages, so that ontological information can serve to put retrieved documents $z$ into context ($p_O(o|z)$) or intermediate LLM reasoning steps ($p_O(o|y_i)$), see snippet A.3.

The fusion operator $\mathcal{F}$ becomes particularly relevant in this case, as now two types of data need be used in equation 2, namely ontological information and document information. In this work, we implement two variations of $\mathcal{F}$, one is a simple concatenation operation, and the other employs an LLM to preprocess and condense the information into a more verbal description of the ontological information, as illustrated in snippet A.3. For specific details and examples, see Appendix A.3.

## 4 OntoGen: Automatic Ontology generation

While OntoRAG offers significant advantages, its use and effectiveness depends on the availability of a high-quality, domain-specific ontology. Creating such ontologies has been traditionally a long and expensive process requiring committees of experts to decide on concepts, relationships, and axioms, see Section 2.1. This limitation can restrict the applicability of OntoRAG to only fields for which ontologies have been already created, excluding new and fast-developing fields—precisely those where such technology could have the highest impact.

To address this challenge, we present OntoGen, a methodology for the automatic creation of ontologies that leverages LLMs to process relevant documents in the field (books, research articles, manuals, etc.) and iteratively curate terms to produce a comprehensive ontology. OntoGen works in three stages that progressively build the ontology from the ground up, as illustrated in Figure 2.

1. **Vocabulary extraction** from the provided texts, identifying key terms and concepts in the domain.

2. Generation of **higher-level categories** for the ontology, providing a broad structure for organizing domain knowledge.

3. Extraction of the **taxonomy** of the ontology, establishing hierarchical relationships between concepts.

This staged approach allows for iterative refinement at each step, ensuring the resulting ontology accurately captures the nuances of the domain while maintaining logical consistency.

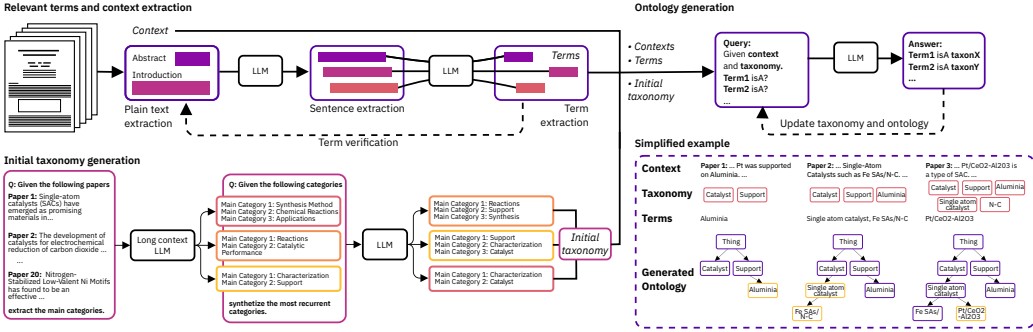

Figure 2: Overview of the proposed Ontology Generation pipeline **OntoGen**.

### 4.1 VOCABULARY EXTRACTION

Vocabulary extraction is a critical first step in our ontology generation process. Our approach leverages Large Language Models (LLMs) to extract domain-specific terms comprehensively while minimizing spurious entries. The process consists of three main steps:

1. **Terms Extraction:** Using LLMs for zero-shot identification of domain-specific terms from sentence-split text, with a grounding check to filter hallucinations.

2. **Acronym Extraction:** Employing a similar LLM-based procedure to identify and pair acronyms with their full terms.

3. **Lemmatization:** Applying lemmatization to group variations of the same term, reducing redundancies in the vocabulary.

Throughout these steps, comprehensiveness is prioritized while maintaining accuracy. Traditional Named Entity Recognition models often struggle with domain generalization (Behr et al., 2023), making our LLM-based approach particularly valuable for emerging or specialized fields. Each extraction step includes a verification process (Appendix A.4) to ensure fidelity to the source text, addressing the challenge of LLM hallucinations. This multi-step process results in a vocabulary that accurately represents the domain knowledge present in the input documents.

### 4.2 CATEGORY EXTRACTION

The first level of the taxonomic hierarchy consists of the main concepts that the ontology will cover. Extracting these high-level categories poses a unique challenge, as individual scientific papers often focus on specific areas within a domain, necessitating a broader perspective across multiple papers.

To address this challenge, we propose the two-step process illustrated in Figure 2b):

1. **Generation:** Conventional LLMs struggle with this task due to limited context windows. We leverage Long-Context LLMs (LCLLMs) (Chen et al., 2023a; Peng et al., 2023; Han et al., 2024; Liu et al., 2023) to overcome this limitation. A random sample of papers is presented to an LCLLM, which is prompted to extract main categories across the given papers. To mitigate the known sensitivity of LCLLMs to input order (Liu et al., 2024a), multiple answers are generated, each with a randomly shuffled order of papers.

2. **Refinement:** Using the answers from the generation step, an LLM is prompted to curate a list of categories based on frequency of occurrence. Self-consistency (SC) techniques (Wang et al., 2022) are applied in this step to ensure categorical consistency across outputs, see Appendix A.4.

The outcome of this process is a seed list of categories that forms the foundation for the rest of the ontology. However, it's important to note that the ultimate correctness and utility of the ontology depend on its intended downstream application. For practical use, some manual effort may be applied at the conclusion of this step in a human-involved alignment phase (see Appendix A.4).

### 4.3 TAXONOMY EXTRACTION

After vocabulary extraction and category generation, the next step is organizing terms into a hierarchical structure using *isA* relationships. We propose a top-down, incremental approach to build a global taxonomy from individual papers.

Given a corpus of $N$ papers $\mathcal{P} = \{P_1, P_2, ..., P_N\}$, with corresponding vocabularies $\mathcal{V} = \{V_1, V_2, ..., V_N\}$, the aim is to construct a taxonomy $\mathcal{T}^{(k)} = \{(s_1, t_1), (s_2, t_2), ..., (s_M, t_M)\}$ at iteration $k$, where each pair $(s_i, t_i) : s_i, t_i \in V_l \in \mathcal{V}$, represents an *isA* relationship.

Our iterative and incremental top-down taxonomy generation process is outlined in Algorithm A.5 and illustrated in Figure 2c. Note that the iterative nature of this process allows for terms that couldn't be placed in earlier iterations to find their proper position in later stages of the taxonomy construction.

## 5 EXPERIMENTS AND RESULTS

To evaluate our methodology, we first gauge the performance of 4 variations of OntoRAG (see Table 5) on 3 standard biomedical Q&A benchmarks: MedMCQA Pal et al. (2022), MedQA Zhang et al. (2018) and MMLU-Med Chen et al. (2023b); Singhal et al. (2023), while using a set of fixed, and expert-curated ontologies in the medical field, namely a biochemistry ontology Aleksander et al. (2023), a general medical term/diagnostic ontology, and the Gene Ontology Consortium (2019) (see Appendix A.2). Questions from these datasets were classified according to medical domains and analyzed accordingly. These experiments aim to evaluate OntoRAG while controlling for the varying components (Section 3).

The results in Table 3 indicate that the use of OntoRAG for medical questions has a similar and sometimes improved performance against baselines. Furthermore, the results in Table 1 demonstrate how, despite modest improvements overall, OntoRAG displays clear advantages in the specific sub-fields where ontologies have been provided, in this work this is the case of *Genetics*, *Anatomy* and to a lesser extent *Microbiology*, see the ontologies used in A.2. It should be noted that the results here serve also to control for performance drops in domains not covered by the given ontologies, which can also be seen in Table 1; however the main aim of OntoRAG is to improve in tasks relevant for scientific discovery.

We argue that the key use-case of OntoRAG is the supplement of scientific discovery in fields for which not enough data exists for an LLM to have developed any internal representation of the field. For well-known fields, there is such a "baked in" ontology from extensive LLM training. Thus, to assess this, the rest of this paper focuses on a rather niche and new field of materials science that fits this description, as data in this field has not yet been ingested and processed by experts into books, much less ontologies.

### 5.1 CASE STUDY: SYNTHESIS OF SINGLE-ATOM CATALYSTS

We further evaluate our approach in the field of Single-Atom Catalysis (SACs), a relatively new field that has gained considerable attention over the last few years Wang et al. (2018). This field is of particular interest due to the relatively small amount of literature published (ca. 800 as of 2023 Liu et al. (2024b)), of which only few are review papers and for which no standard books have yet been written. Clearly, no expert-curated ontology exists as of now.

#### 5.1.1 ONTOLOGY GENERATION ANALYSIS

Generating an ontology for SACs presented unique challenges due to the field's novelty. We collected a corpus of 500 recent research papers on SACs as our data source, see Appendix A.5.1, and they were processed as illustrated in Section 4 using two different LLMs: Claude-3.5-sonnet (Anthropic (2023)) and Llama-3.1-70B (Touvron et al. (2023)), in order to assess the model-dependence of the generated ontologies.

Displayed in Figure 3 are different descriptors for the ontologies, for the two LLMs and across 5 cycles of OntoGen. The results show a clear difference between the ontologies generated by each LLM, in terms of every variable measured. In general, Claude seems to not modify it's ontology beyond step 2, while Llama continuously adds nodes and relationships until iteration 4. In terms of topology, Claude's ontology seems to be more flat or linear, with branching factors lying close to 1 while Llama's are almost twice in average.

These results are not conclusive of any particular advantage of any model over the other, and more measures of quality should be set in place in order to evaluate the method.

#### 5.1.2 ONTORAG FOR SYNTHESIS OF SACS

Despite their importance and potential, a key bottleneck in the discovery process of SACs is their synthesis Fang et al. (2023). Some work has already been reported on the use of Language Models for modeling syntheses of SACs Suvarna et al. (2023) applied to data extraction and curation. We thus decided to tackle the challenge of directly predicting synthetic procedures for SACs specified in free-form natural language by users.

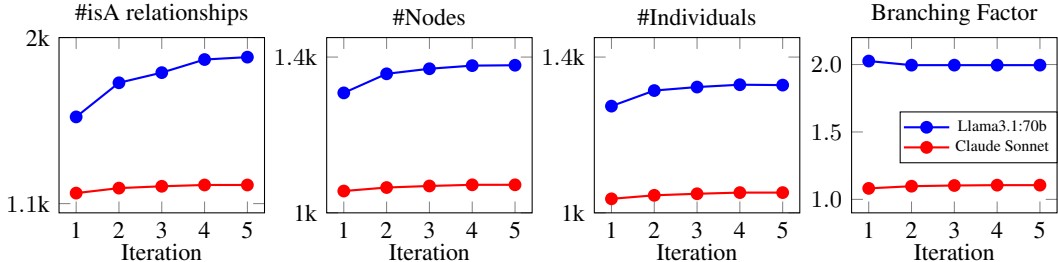

Figure 3: Number of nodes, *isA* realtionships, individuals and average branching factor of the generated SACs ontologies over iterations with Llama3.1:70b (blue) and Claude Sonnet 3.5 (red).

**Benchmark**  SACBench is a benchmark to evaluate the ability of RAG systems at predicting synthesis procedures for given SACs. It was generated and manually curated from a corpus of research papers in the field, and each datapoint corresponds to a single SAC with its corresponding synthesis procedure, processed into structured steps using Llama 70B (Touvron et al., 2023). The benchmark contains 51 SAC-requests extracted from 28 papers, and covers a diverse space of metals, chemicals, and synthesis methods, making it a challenging dataset to test chemistry capabilities of LLMs and RAG systems. In addition, the benchmark includes a series of field-relevant metrics, such as completeness and accuracy of the procedure, along with precision and accuracy in the use of metals and chemicals. Refer to Appendix A.5 for more details.

We evaluate OntoRAG variations against two baselines: ZeroShot and RAG with Chain-of-Thought Wei et al. (2022), using 3 LLMs from different providers: Llama-3.1-8B Dubey et al. (2024), GPT-4o-mini 12, and Claude-3.5-Sonnet 14. Results in Figure 4 and Appendix 7 show a clear progression of performance from the smallest model (Llama) to the largest (Claude 3.5) across most metrics. OntoRAG is indeed the best-performing method across all *metal* and *chemicals* metrics as compared to ZeroShot and CoT, with the *support* metrics improving only for Claude-3.5. [these metrics are important]

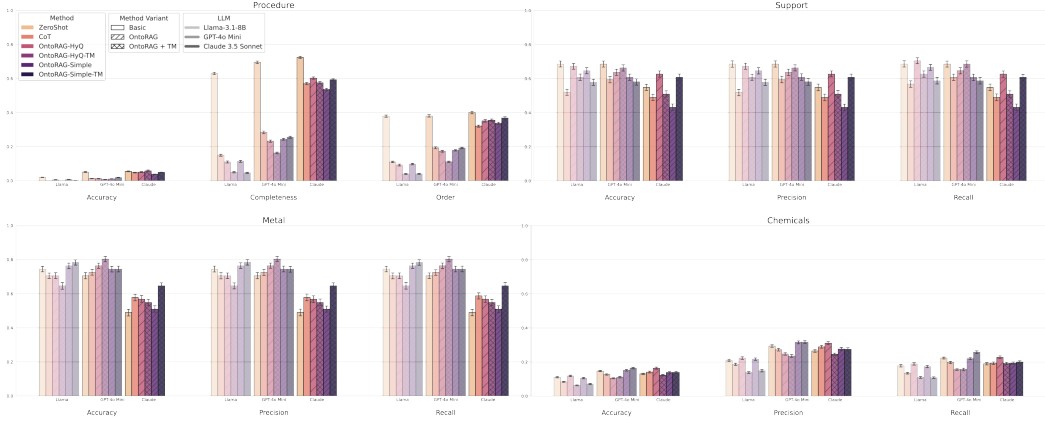

Figure 4: RAG accuracies for each method (see Table 5), for each of 3 LLMs: Claude-3.5-Sonnet Anthropic (2023), gpt-4o-mini OpenAI (2023), Llama-3.1-8B Touvron et al. (2023). **Accuracy** measures that the correct steps are predicted at the correct order, **Completeness** measures that all the synthetic steps are present in the generated procedure, and **Order** measures that steps are mentioned in the correct order.

Notably, ZeroShot is the best method at the *procedure* metrics, substantially above any variation of OntoRAG and CoT. As shown in Figure A.8, there is a clear difference in response length distributions between ZeroShot and all other methods, and this is consistent across LLMs. Because the *procedure* metrics depend heavily on the specificity, length and detail provided by the LLM (see Appendix

A.5), the drop in performance is a direct cause of generally shorter responses. Additionally, these performance gaps correlate well with the accuracy differences found in Figure 7.a.

One key observation when generating ontologies for fields where no ground truth ontology has been created is that quality can only be (scalably) measured on the downstream performance on some downstream application. This is the case of our benchmark, and so we analyze the differences in OntoRAG performance when generating the ontologies with different LLMs, Claude-3.5-Sonnet and Llama-3.1-70B in this case (Section 5.1.2). Tables 4 to 6 show that, while there are variations in performance across different LLMs and OntoRAG methods, our methods perform similarly or with slight improvement over baseline measures. Notably, Claude paired with OntoRAG displays significant improvement over baseline metrics.

## 6 DISCUSSION

We present OntoRAG, a method for performing RAG using context from field-specific ontologies. The method is complemented by OntoGen, a method for automatic generation of ontologies from a corpus of papers, which allows the use of OntoRAG even for fields where no ontology has been defined. For evaluation we select the task of planning synthesis procedures for Single-Atom Catalysts, and present a novel, diverse and challenging benchmark.

Our experimentation with the pipeline on both general domain benchmarks and focused, narrow benchmarks indicate the variability in performance depending on the questions asked, the ontologies that are utilized, and the language model employed for principle tasks. This variability represents the importance of the knowledge used to generate the ontology, and should be carefully selected by humans to ensure maximum performance. Furthermore, the success of OntoRAG depends on the quality and relevance of the ontology utilized.

Our contributions highlight the potential for using LLMs as not only a method of extracting knowledge for downstream applications, but also as a means of leveraging text-based knowledge for the structured extraction and formalization of domain specific concepts in the form of an ontology. We hope that the ideas, benchmarks and analyses presented in this paper will contribute to the construction of more reliable, accurate and interpretable AI systems for accelerating scientific discovery across domains.

## 7 LIMITATIONS AND FUTURE WORK

The results of our work demonstrate some fundamental limitations. The principal limitation is the computational cost of generating the ontology, and the variability that arises from different knowledge sources and language model. The variability in the OntoGen pipeline highlights the ongoing importance of human expertise in ontology development. We also recognize that, while the integration of ontological information may improve QA performance on question answering tasks in particular domains, there is no direct improvement over the conventional RAG system on a benchmark-wide scale. We attribute this to limitations in the ontologies utilized rather than the pipeline itself. More studies are needed to decide conclusively on the effect that relevant ontological information has on the reasoning process and question answering accuracy of an LLM.

In future work, we aim to further integrate ontologies into the RAG pipeline. The core focus of future work will be direct comparison of performance on bench marks between one shot LLM, LLM- RAG incorporation, and LLM-RAG-ontology integration. Furthermore, we future studies will integrate open-ended question answering so that the reasoning of each system can be more thoroughly evaluated. We also interned to utilize more specific and human made ontologies of niche domains (such as single-atom catalysis) for direct analysis of ontologies generated by the OntoGen pipeline, in addition to the downstream integration into the OntoRAG pipeline.

AUTHOR CONTRIBUTIONS

ACKNOWLEDGMENTS

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
