# A  APPENDIX

## A.1  CONTROL EXPERIMENTS ON BIOMEDICAL BENCHMARKS

### A.1.1  PERFORMANCE AND ANALYSIS

| Method | TM | MedMCQA | MedQA | MMLU-Med |
|---|---|---|---|---|
| ZeroShot | ✗ | 62.06 | 67.16 | 80.06 |
| CoT | ✗ | 60.91 | **69.99** | 76.70 |
| OntoRAG-simple | ✗ | **64.12** | 68.34 | 79.26 |
| | ✓ | 61.80 | 68.11 | 80.01 |
| OntoRAG-HyA | ✗ | **64.04** | 67.64 | 79.96 |
| | ✓ | 62.13 | **69.36** | **80.65** |

Table 1: Performance comparison of methods on 3 biomedical benchmarks. TM denotes "*translation module*", refering to a variation of the fusion operator $\mathcal{F}$ in which an LLM translates ontological context into natural language.

### A.1.2  EFFECTS OF ONTOLOGICAL RELEVANCE.

We hypothesize that weak performance in some areas when using OntoRAG might be due to vocabulary discrepancies as an effect of decreased ontological relevance. The assess this, we conduct an analysis where for each question in a given benchmark, the number of retrieved concepts from an ontology is computed, and the mean across the benchmark is correlated to performance (accuracy), for a given method. That is, each ontorag variation contributes one point to the correlation analysis. The goal is to determine whether high ontological relevance correlates with higher accuracy.

The results in Table 2 indicate an overall positive and usually strong correlation between ontological relevance and downstream performance.

| Benchmark | Correlation |
|---|---|
| MedQA | 0.7852 |
| MMLU-Med | 0.7506 |
| MedMCQA | 0.1018 |

Table 2: Correlation values for different benchmarks

## A.2  MEDICAL ONTOLOGIES

We first evaluate our methodology by first gauging its performance on a well known LLM question and answer (QA) benchmark, Multi-Subject Multi-Choice Dataset for Medical domain (MedMCQA) (Pal et al., 2022). This is a popular benchmark for evaluating LLM performance on multiple choice questions from various areas in the medical domain. Questions from this dataset were first divided based on their medical domain (dentistry, pediatrics, etc.)  which then guided the selection of ontologies to place into the OntoRAG pipeline. The selected ontologies were limited to a biochemical ontology (https://bioportal.bioontology.org/ontologies/REX ) a general medical term/ diagnostic ontology (https://bioportal.bioontology.org/ontologies/SNOMEDCT), and the widely-used gene ontology (GO) Aleksander et al. (2023) in an attempt to cover most of the concepts present in the QA dataset. These ontologies were also chosen due to their public availability and their professional quality. The benchmark was was curated to only include concepts that appear within the utilized ontologies. The final dataset contained around 4000 questions with the number of questions ranging from 27 to 400 for each medical domain.  As with the results presented in the main document, the OntoRAG system offers similar or improved performance over the baseline zero-shot and CoT methods, with a significant improvements in the areas of genetics, anatomy, and microbiology. These improvements correlate with the fact that we used ontologies most relevant to these fields.

| No. Entries | Question Class | ZeroShot | CoT | OntoRAG |
|---|---|---|---|---|
| 405 | Unknown | **0.83** | 0.78 | 0.82 |
| 311 | Biochemistry | 0.81 | 0.78 | **0.83** |
| 283 | Physiology | **0.82** | 0.79 | **0.82** |
| 130 | Medicine | **0.88** | 0.83 | 0.86 |
| 92 | Preventive Medicine | **0.75** | 0.65 | 0.71 |
| 88 | Microbiology | 0.58 | 0.57 | **0.61** |
| 80 | Gynaecology & Obstetrics | **0.82** | 0.78 | **0.82** |
| 77 | Anatomy | 0.77 | 0.77 | **0.91** |
| 72 | Pharmacology | 0.78 | **0.79** | 0.76 |
| 68 | Pediatrics | 0.85 | **0.87** | 0.85 |
| 49 | Psychiatry | 0.73 | **0.76** | 0.73 |
| 33 | Surgery | **0.73** | 0.67 | 0.61 |
| 23 | Dental | **0.74** | 0.65 | **0.74** |
| 18 | Genetics | 0.83 | 0.78 | **0.89** |
| 18 | Orthopaedics | **0.83** | 0.67 | **0.83** |
| 16 | Neurology | **0.88** | 0.81 | 0.81 |

Table 3: Accuracy of OntoRAG against baselines on MMLU-Med, by question class. The table shows the accuracy of each method by type of question. OntoRAG-HyA-TM was used here.

### A.3 ONTORAG DETAILS

OntoRAG is implemented using the DSPy library Khattab et al. (2023). The library abstracts the interface with an LLM into Signatures and Modules. The Signatures abstract the prompting of the LLM into classes with Input and Output properties, while the Modules define the flow of information that the pipeline implements.

The below Module is defined as the OntoRAG base module, and defines some standard routines used in every other sub-module used in this work.

Figure 5: OntoRAG implementations used in this work. Only *Simple* and *HyQ* are shown here. These represent variations in the retrieval type (i.e. direct or hypothetical answer). Variations in the fusion operator F are defined as part of the BaseOntoRAG class, see Appendix A.3.

```python
class ORAG_Simple(BaseOntoRAG):
    """Simple Ontorag"""
    def forward(self, q: str):
        ctxt = self.retr(q)
        answer = self.predictor(
            question=q,
            context=context
        )
        return answer
```

OntoRAG Simple

```python
class ORAG_HyA(BaseOntoRAG):
    """Ontorag with Hypot. answer
    ↪ """
    def forward(self, q: str):
        # Hypothetical answer
        ctxt0 = self.retr(q)
        hans = self.hya(
            question=q,
            context=ctxt0
        )
        # Query concepts in HyA
        ctxt1 = self.retr(
            hans.answer
        )
        answer = self.predictor(
            question=q,
            context=ctxt1
        )
        return answer
```

OntoRAG-HyA

---

**Algorithm 1** OntoRAG base class.

---

```python
class BaseOntoRAG(dspy.Module):
    retriever: dspy.Retrieve
    ontoretriever: OntoRetriever

    def forward(self, query: str) -> dspy.Prediction:
        """Forward pass of the OntoRAG pipeline."""
        pass

    def retrieve(self, query: str, ctxt_doc: str|None) -> str:
        """Retrieve and format."""
        ctxt_doc, ctxt_onto = "", ""

        if ctxt_doc is None:
            ctxt_dict = self.retrieve_doc(query)
            ctxt_doc = self.format_context(ctxt_dict)

        if self.ontoretriever.ontology.ontologies:
            ctxt_ontoj = self.ontoretriever(query)
            ctxt_onto = self.format_onto_context(ctxt_ontoj)

        ctxt = self.fuse_contexts(ctxt_doc, ctxt_onto)
        return ctxt

    def format_context(self, context: List[Dict]) -> str:
        """Format context."""
        contexts = [p["text"] for c in context for p in c["passages"]]
        return "\n".join(deduplicate(contexts))

    def format_onto_context(self, context: List[Dict]) -> str:
        """Format ontology context."""
        return json.dumps(context, indent=2)

    def fuse_contexts(self, ctxt_doc: str, ctxt_onto: str) -> str:
        """Fuse document and ontology contexts."""
        return ctxt_doc + ctxt_onto
```

---

A specific implementation of OntoRAG looks as follows: First, a Signature is defined, where inputs and outputs are defined.

The Modules are written to handle the inputs in the Signature, and to produce the outputs.

### A.3.1 ONTOLOGY RETRIEVAL OPERATOR

The operator $\mathcal{O}$ defined in eq. 2, works by first extracting concepts from a statement $s$ and returning the most similar ontological concepts $\{o\}$ in the ontology. The concepts are retrieved by 1. extracting concepts from the input query, and 2. retrieving ontological context from each of those concepts. The complete ontology retrieval pipeline is illustrated in pseudo-code 4.

In our implementation, retrieval works by extracting concepts using the spacy "en_core_web_sm" parser. The pipeline then searches in the loaded ontology, and if found retrieves the parents, children, as well as the definition, if any.

**Algorithm 2** MedQnA: Medical Question Answering Signature

```
class MedQnA(dspy.Signature):
    """Answer a question with a detailed response based on the
    given context. If the context is not relevant or there is no
    context, answer based on
    your knowledge."""

    context: str = dspy.InputField(
        desc="Context: This information shows the relationships between
        relevant concepts:"
    )
    question: str = dspy.InputField(
        desc="Here is the question you need to answer:"
    )
    reasoning: str = dspy.OutputField(
        desc="Reasoning: Let's think step by step in order to ${reasoning}"
    )
    choice_answer: str = dspy.OutputField(desc="Answer: ${answer}")
```

**Algorithm 3** SimpleORAG: Simple Ontology-enhanced Retrieval-Augmented Generation

```
class SimpleORAG(BaseOntoRAG):

    def __init__(
        self,
        ontology: Union[str, OntoRetriever],
        context: None|str,
    ):
        super().__init__()
        self.predictor = dspy.Predict(MedQnA)
        if isinstance(ontology, str):
            self.ontoretriever = OntoRetriever(ontology_path=ontology)
        else:
            self.ontoretriever = ontology

    def forward(self, qprompt: str) -> dspy.Prediction:
        context = self.retrieve(qprompt)
        answer = self.predictor(question=qprompt, context=context)
        return answer
```

### A.3.2 WORKING EXAMPLE OF ONTORAG.

Here we need to show an example of a variation of ontorag.

## A.4 ONTOGEN DETAILS

### A.4.1 SELF CONSISTENCY

The improvement of LLMs' capabilities to generate high-quality, hallucination-free answers is currently a highly active area of research. Many generic methods have been proposed that improve LLMs outputs without training data, fine-tuning or reinforcement learning, which includes, among others, self-consistency Wang et al. (2022), debating LLMs Du et al. (2023), and self-refinement Madaan et al. (2024). Research by Huang et al. Huang et al. (2023) demonstrates that self-consistency offers competitive results while being more computationally efficient compared to other methods. Therefore, in this work, self-consistency is used to improve the quality of answers from a LLM. As utilized in our approach, self-consistency can be defined as:

**Algorithm 4** Retrieval of ontological context

```
 1: procedure PROCESSQUERY(query)
 2:     recognizedConcepts ← RecognizeConcepts(query)
 3:     output ← ∅
 4:     for each ontology, concepts in recognizedConcepts do
 5:         for each concept in concepts do
 6:             context ← GetOntologicalContext(concept, ontology)
 7:             output[ontology][concept] ← context
        return output
 8: procedure RECOGNIZECONCEPTS(text)
 9:     doc ← NLP(text)
10:     recognizedConcepts ← ∅
11:     for each token in doc do
12:         if token matches any ontology pattern then
13:             concept ← token.text
14:             ontology ← DetermineOntology(concept)
15:             recognizedConcepts[ontology].add(concept)
        return recognizedConcepts
16: procedure GETONTOLOGICALCONTEXT(concept, ontology)
17:     class ← ontology.search(label = concept)
18:     context ← {
19:       "label" : class.label,
20:       "definition" : class.definition,
21:       "parents" : class.superclasses(),
22:       "children" : class.subclasses()
23:     } return context
```

**Definition A.1** *Let $a_1, a_2, ..., a_n \in \mathbb{A}$ be the answers to a given prompt $p$ generated by a LLM, and $r_i$ the set of tokens generated before the answer $a_i$.*

*Self-Consistency (SC) applies a marginalization over $r_i$ by taking the majority vote of the answers $a_i$, i.e. $a = \arg\max_{a_i} \sum_{j=1}^{n} \mathbb{1}(a_i = a_j)$, thus giving as a final answer the most "consistent" answer generated by the LLM.*

It is important to note that self-consistency was initially proposed to enhance Chain of Thought (CoT) reasoning Wei et al. (2022) in LLMs Wang et al. (2022), to improve performance on generalized problem-solving tasks. In our work, we leverage the generalizability of self-consistency to improve the quality of our knowledge schemas reconstruction.

### A.4.2   VOCABULARY EXTRACTION

After each iteration with the LLM, when it has extracted a list of concepts, a verification step is performed that consists of performing a string search of each of the list terms, in the original sentence. Terms pass this filter only if they are contained in the original sentence. With this process, we terms that originate as a result of hallucinations from the LLM used.

### A.4.3   CATEGORIES GENERATION

During the *refinement* step, the LLM is prompted to curate a list of the most frequent categories extracted from the previous step. SC is applied here by generating many answers from the same prompt, and taking the majority vote of the categories extracted. While this provides a more robust list of categories, it is important to note that the correctness of an ontology is dependent on the downstream application it is intended for. Therefore, human involvement may be required in this step to select or exclude certain categories in order to align it with the downstream application. The final list of categories is then used as a seed for extracting the entire taxonomy, making it crucial to ensure the list is of high quality.

In the case of SACs ontology, the generated list of categories, obtained by majority voting was: *Characterization, Physical properties, Synthesis methods, Reaction mechanisms, Structure, Applications, Reactions* and *Support*. The manual curation performed in this step involved selecting the following additional categories from the pool of generated categories, so as to make the ontology more aligned with our chemistry knowledge: *Catalytic performance, Preparation methods, Theory and modelling,* and *Materials*.

### A.4.4 ALGORITHM FOR TAXONOMY GENERATION

---

**Algorithm 5** Iterative and Incremental Top-Down Taxonomy Generation

---

**Input:** Papers $\mathcal{P}$, Vocabulary $\mathcal{V}$, Initial Taxonomy $\mathcal{T}^{(0)}$
**Output:** Reconstructed Taxonomy after $K$ iterations $\mathcal{T}^{(K)}$

1: **for** $k = 1, \ldots, K$ **do**
2:     $\mathcal{T}^{(k)} \leftarrow \mathcal{T}^{(k-1)}$
3:     **for** $P_i \in \mathcal{P}$ **do**
4:         $R_i \leftarrow$ query_relationships$(P_i, V_i, \mathcal{T}^{(k)})$
5:         **for** $(s, t) \in R_i$ **do**
6:             **if** is_valid$((s, t), \mathcal{T}^{(k)})$ **then**
7:                 $\mathcal{T}^{(k)} \leftarrow \mathcal{T}^{(k)} \cup \{(s, t)\}$
8: **return** $\mathcal{T}^{(K)}$

---

Where,

- `query_relationships`: Extracts *isA* relationships $(s, t)$ from paper $P_i$, where $s \in \mathcal{C}(\mathcal{T}^{(k)})$ is a term in the current taxonomy $\mathcal{T}^{(k)}$, and $t \in V_i$. This function aims to place each term into the existing taxonomy, potentially returning multiple relationships per term.

- `is_valid`: Ensures no loops are created in the taxonomy when inserting a new relationship.

In our implementation, `query_relationships` utilizes an LLM prompted with the paper content, the current taxonomy terms, and the vocabulary to be queried. An example prompt and response can be found in Appendix A.6. To enhance the quality of the generated taxonomy and reduce hallucinations, SC is applied in this step by generating multiple answers from the same prompt and taking the majority voting as the final answer.

### A.4.5 EXPERT EVALUATION

In order to evaluate the quality of the generated ontology, a panel of two experts was assembled to assess the taxonomical relationships. The experts were tasked with randomly sampling relationships from various iterations of the ontology and determining whether each sampled relationship was correct according to the context provided for such relationship, in this case, the corresponding paper. According to the experts, on average at least 64.5% of the sampled relationships were considered correct. While this indicates a majority of accurate relationships, it also suggests room for improvement in the ontology generation process. Upon analysis of the incorrect relationships, the experts identified as potential improvements the removal of semantically similar concepts, which might appear repeated in different parts of the structure, and the need to provide a more specific context for the relationships, in order to reduce ambiguity.

### A.4.6 SACS ONTOLOGY EXAMPLE

To provide a concrete example of how the ontology is able to capture meaningful relationships, below two examples are provided corresponding to the *synthesis methods* (left) and *CO2 reduction reactions* (right) branches for both the ontologies generated with Claude 3.5 Sonnet and Llama3.1:70b. Here it can be seen that both ontologies are able to capture meaningful synthesis methods for SACs that appear in the literature. It can be seen that, generally there is an agreement in the synthesis methods identified in both ontologies. It can be highlighted, however, that the Llama-generated ontology contains a larger number of false-positive synthesis methods (e.g. *Methodology, Synthesis, Strategies*), which

explains the larger number of terms included in this ontology. Regarding the *CO2 reduction* branch, one can notice that each ontology contains semantically similar terms (e.g. *Carbon dioxide reduction reaction* and *CO2 reduction reaction*). While this does not affect the downstream performance of the ontology, it creates unnecessary redundancies in the structure. Additionally, it can be seen that, in the Llama-generated ontology, *CO2 reduction* has not been classified as a separate branch, but instead, it is contained inside the *Reactions* branch, without this being necessarily incorrect. Finally, as it happened with the *synthesis methods* branch, the Llama-generated ontology contains evident false-positives (e.g. *CO2 molecules, dioxide*), which did not appear in the Claude-generated ontology.

**Example SACs Ontology (Claude 3.5 Sonnet)**

```
Thing
└── Synthesis methods
    ├── Catalyst synthetic strategies
    ├── Two-step approach
    ├── Ni-TAPc anchoring strategies
    ├── Pyrolysis procedure
    ├── Bimodal template based synthesis strategies
    ├── Multistep pyrolysis process
    ├── Multistep pyrolysis method
    ├── Wet chemistry methods
    ├── Pyrolysis
    ├── Atomic layer deposition
    ├── Pyrolysis process
    ├── NH3 atmosphere annealing
    ├── Co precipitation
    ├── Annealing
    ├── Lyophilization
    ├── Galvanic replacement reaction
    ├── Synthetic process
    ├── Incipient wetness impregnation
    ├── Synthesis approach
    ├── Silica templating
    ├── Synthetic approaches
    ├── Synthesis
    ├── Synthesis condition
    ├── Heteroatom doped
    ├── Reduction temperature
    ├── Hydrothermal ethanol reduction method
    ├── High-temperature pyrolysis
    ├── Immobilization via functional group
    ├── Dendrimer encapsulation
    ├── Hydrothermal treatment
    ├── Impregnation methods
    ├── Wet impregnation
    ├── Sol-gel approach
    ├── Self-assembly route
    ├── Synthetic strategies
    └── High-temperature self-assembly route
```

```
Thing
└── Reactions
    └── CO2 reduction
        ├── Electrochemical carbon dioxide reduction
        ├── Carbon dioxide reduction reaction
        ├── CO2 reduction reaction (CO2RR)
        ├── Electrochemical CO2-to-CO conversion
        ├── Electrochemical CO2 reduction reaction (CO2RR)
        ├── CO2 conversion
        ├── eCO2RR
        ├── CO2 electroreduction
        ├── Photocatalytic CO2 conversion
        ├── Photocatalytic CO2 reduction reaction
        ├── CO2 to CO conversion
        ├── Photocatalytic reduction
        ├── CO2 photoreduction
        ├── Catalytic CO2 conversion
        ├── CO2 hydrogenation
        └── Electroreduction
```

**Example SACs Ontology (Llama 3.1:70b)**

```
Thing
└── Synthesis methods
    ├── Catalyst synthetic strategies
    ├── Nanoconfined ILs strategy
    ├── Solid liquid interface engineering
    ├── Confinement
    ├── Synthesis
    ├── Strategies
    ├── Postprocessing solution treatments
    ├── Acidic leaching
    ├── Sol-gel approach
    ├── Incipient wetness impregnation
    ├── Annealing
    ├── Lyophilization
    ├── Galvanic replacement reaction
    ├── Atomic layer deposition
    ├── Co-precipitation
    ├── Alloying
    ├── Synthetic process
    ├── NH3 atmosphere annealing
    ├── Hydrothermal treatment
    ├── Oxychlorination
    ├── Iodo hydrocarbon treatment
    ├── NO/CO treatment
    ├── Dendrimer encapsulation
    ├── Repetitive oxidation and reduction
    ├── Immobilization via functional group
    ├── Pyrolysis procedure
    ├── Bimodal template based synthesis strategies
```

```
Thing
└── Reactions
    ├── CO2 molecules
    ├── CO2 reduction
    ├── Electrochemical CO2 reduction reaction (CO2RR)
    ├── Carbon dioxide
    ├── CO2 emissions
    ├── CO2 reduction reaction (CO2RR)
    ├── Anthropogenic CO2 emissions
    ├── Carbon dioxide reduction reaction
    ├── Electrochemical carbon dioxide reduction
    ├── Photocatalytic CO2 reduction reaction
    ├── CO2 to CO conversion
    ├── dioxide
    ├── eCO2RR
    ├── CO2 electroreduction
    ├── CO2 photoreduction
    ├── CO2 conversion
    ├── CO2 activation
    ├── Electrochemical CO2 to CO conversion
    ├── <remaining omitted for clarity>
```

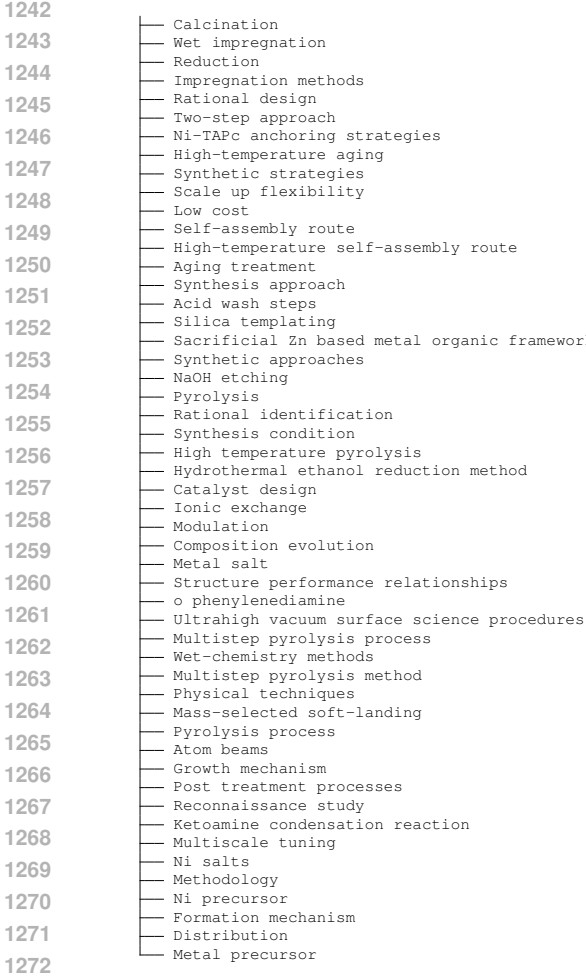

```
├── Calcination
├── Wet impregnation
├── Reduction
├── Impregnation methods
├── Rational design
├── Two-step approach
├── Ni-TAPc anchoring strategies
├── High-temperature aging
├── Synthetic strategies
├── Scale up flexibility
├── Low cost
├── Self-assembly route
├── High-temperature self-assembly route
├── Aging treatment
├── Synthesis approach
├── Acid wash steps
├── Silica templating
├── Sacrificial Zn based metal organic framework
├── Synthetic approaches
├── NaOH etching
├── Pyrolysis
├── Rational identification
├── Synthesis condition
├── High temperature pyrolysis
├── Hydrothermal ethanol reduction method
├── Catalyst design
├── Ionic exchange
├── Modulation
├── Composition evolution
├── Metal salt
├── Structure performance relationships
├── o phenylenediamine
├── Ultrahigh vacuum surface science procedures
├── Multistep pyrolysis process
├── Wet-chemistry methods
├── Multistep pyrolysis method
├── Physical techniques
├── Mass-selected soft-landing
├── Pyrolysis process
├── Atom beams
├── Growth mechanism
├── Post treatment processes
├── Reconnaissance study
├── Ketoamine condensation reaction
├── Multiscale tuning
├── Ni salts
├── Methodology
├── Ni precursor
├── Formation mechanism
├── Distribution
└── Metal precursor
```

## A.5    SACBENCH: BENCHMARK FOR SAC SYNTHESIS PROCEDURES

SACBench is a comprehensive benchmark designed to evaluate the performance of systems that generate experimental procedures for the synthesis of Single-Atom Catalysts (SACs). The benchmark consists of 50 input-output pairs, where the input specifies a desired SAC and the output is the correct synthesis procedure.

The evaluation metrics used aim to assess the validity and correctness of a generated synthesis suggestion, in chemically meaningful terms.

Some metrics include:

1. Procedure Accuracy: Measures the overall correctness of the generated procedure.

2. Procedure Completeness: Assesses how comprehensive the generated procedure is compared to the reference.

3. Procedure Order: Evaluates the correct sequencing of steps in the generated procedure.

4. Chemical Identification: Includes recall, precision, F1 score, and accuracy for identifying correct chemicals in the procedure.

5. Metal Identification: Measures recall, precision, F1 score, and accuracy for correctly identifying the metal component of the SAC.

6. Support Identification: Evaluates recall, precision, F1 score, and accuracy for correctly identifying the support material in the SAC synthesis.

Figure 6 shows some general statistics about the test dataset, and the co occurrences between different variables.

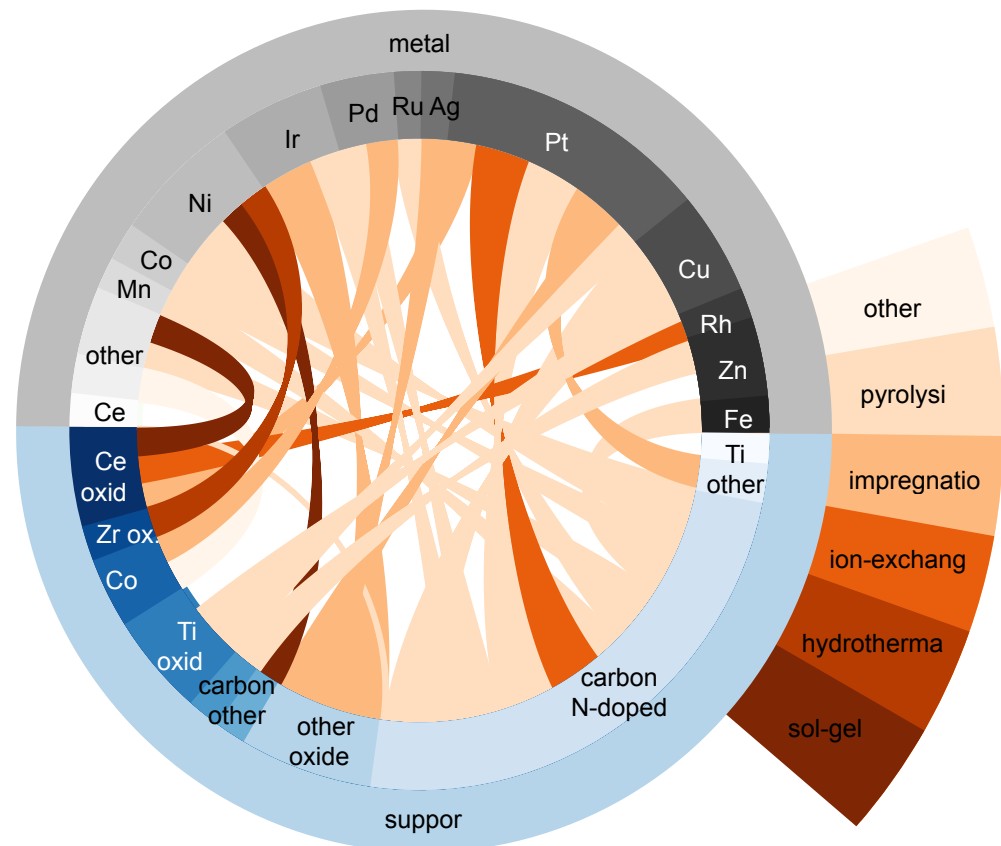

Figure 6: Descriptive statistics of the benchmark created for this work.

### A.5.1 SAC RESEARCH PAPERS CORPUS

The corpus of 500 recent research papers on Single-Atom Catalysts (SACs) used for ontology generation includes publications from top journals in catalysis and materials science from the past 5 years. The papers cover various aspects of SACs, including synthesis methods, characterization techniques, and applications.

The research papers were obtained from Wiley Journals through Wiley's official API (Wiley-API (2024)).

### A.6 PROMPT EXAMPLE

Here's an example prompt used in the `query_relationships` function for taxonomy extraction:

```
Given the following paper content, current taxonomy terms, and vocabulary
    ↪  to be queried, please identify 'isA' relationships between terms
    ↪ in the vocabulary and terms in the current taxonomy. Ensure that
    ↪ each relationship is supported by evidence from the paper content.

Paper content:
In the field of catalysis, single-atom catalysts represent a specialized
    ↪ form of catalysts, emerging from the parent concept of a catalyst
    ↪ but with isolated active sites at the atomic level. Their creation
```

```
    ↪ often involves various synthesis methods, with wet impregnation
    ↪ being a common technique to distribute the active metal atoms
    ↪ evenly on a support. Once synthesized, these catalysts can be
    ↪ characterized using X-ray absorption spectroscopy.

Current taxonomy terms:
- Reactions
- Catalyst
- Materials
- Synthesis method
- Characterization technique
- Preparation method

Vocabulary to be queried:
- Single-atom catalyst
- Wet impregnation
- X-ray absorption spectroscopy

Please format your response as a list of relationships in the form (
    ↪ parent_term, child_term), where parent_term is from the current
    ↪ taxonomy and child_term is from the vocabulary to be queried."
```
Listing 1: Prompt Example

```
Here is the list of relationships:

(Catalyst, Single-atom catalyst)
(Synthesis method, Wet impregnation)
(Characterization technique, X-ray absorption spectroscopy)
```
Listing 2: Response Example

## A.7 DOWNSTREAM EVALUATION OF ONTOLOGIES

Evaluating the quality of generated ontologies requires either careful expert evaluation, typically involving committees of experts in the field Keet (2018), or downstream applications that use them as an integral part of the pipeline and provide quantitative result of some sort.

In our work, we opt for the downstream application on SAC Synthesis to compare two SAC ontologies generated with OntoGen, using LLMs of different capacity, namely Claude-3.5-Sonnet, and Llama-3.1-70B. We compare two variants of OntoRAG-simple: with and without a Translation Module. Additionally we include the results of the ZeroShot and CoT baselines for comparison. All the results in Tables 4 to 6 are results with gpt-4o-mini as LLM. The metrics used are defined in Appendix A.5.

Table 4: ZeroShot (Baseline)

| ontology | completeness | Procedure order | accuracy | Chemicals accuracy | Metal accuracy | Support accuracy |
|---|---|---|---|---|---|---|
| Claude | 0.725011 | 0.400722 | 0.055564 | 0.130818 | 0.490196 | 0.549020 |
| Llama | 0.725011 | 0.400722 | 0.055564 | 0.130818 | 0.490196 | 0.549020 |

Table 5: CoT (Baseline)

| ontology | completeness | procedure order | accuracy | chemicals accuracy | metal accuracy | support accuracy |
|---|---|---|---|---|---|---|
| Claude | 0.570561 | 0.321268 | 0.048420 | 0.141569 | 0.578431 | 0.490196 |
| Llama | 0.570561 | 0.321268 | 0.048420 | 0.141569 | 0.578431 | 0.490196 |

## A.8 SACBENCH RESULTS & ANALYSIS

Table 6: OntoRAG-simple

| ontology | procedure | | | chemicals | metal | support |
| | completeness | order | accuracy | accuracy | accuracy | accuracy |
|---|---|---|---|---|---|---|
| Claude | **0.577304** | 0.330314 | 0.044630 | 0.130324 | **0.607843** | **0.490196** |
| Llama | 0.536076 | 0.337008 | 0.038061 | **0.138353** | 0.509804 | 0.431373 |

Table 7: OntoRAG-simple-tm

| ontology | procedure | | | chemicals | metal | support |
| | completeness | order | accuracy | accuracy | accuracy | accuracy |
|---|---|---|---|---|---|---|
| Claude | **0.613198** | 0.364592 | 0.049093 | 0.132388 | **0.705882** | 0.568627 |
| Llama | 0.593519 | 0.369136 | 0.049502 | 0.138899 | 0.647059 | **0.607843** |

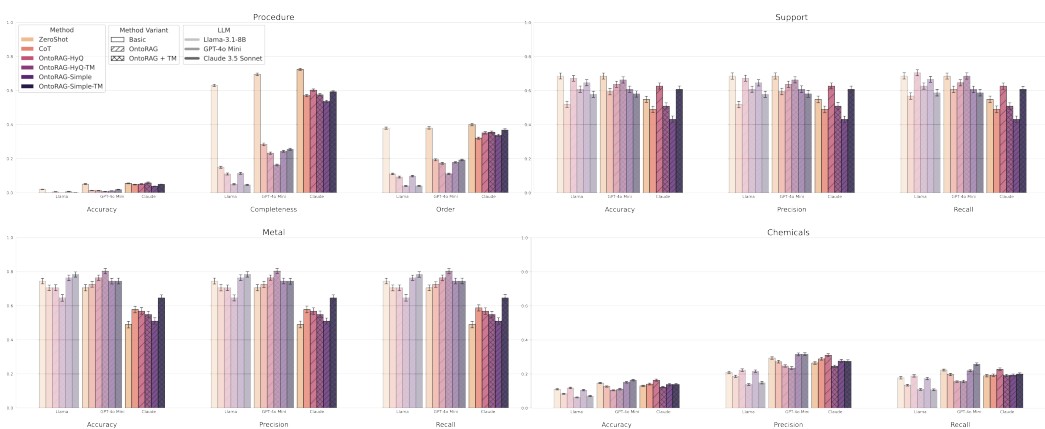

Figure 7: Complete results of multiple methods, and LLMs, on multiple metrics of the SACBench benchmark.

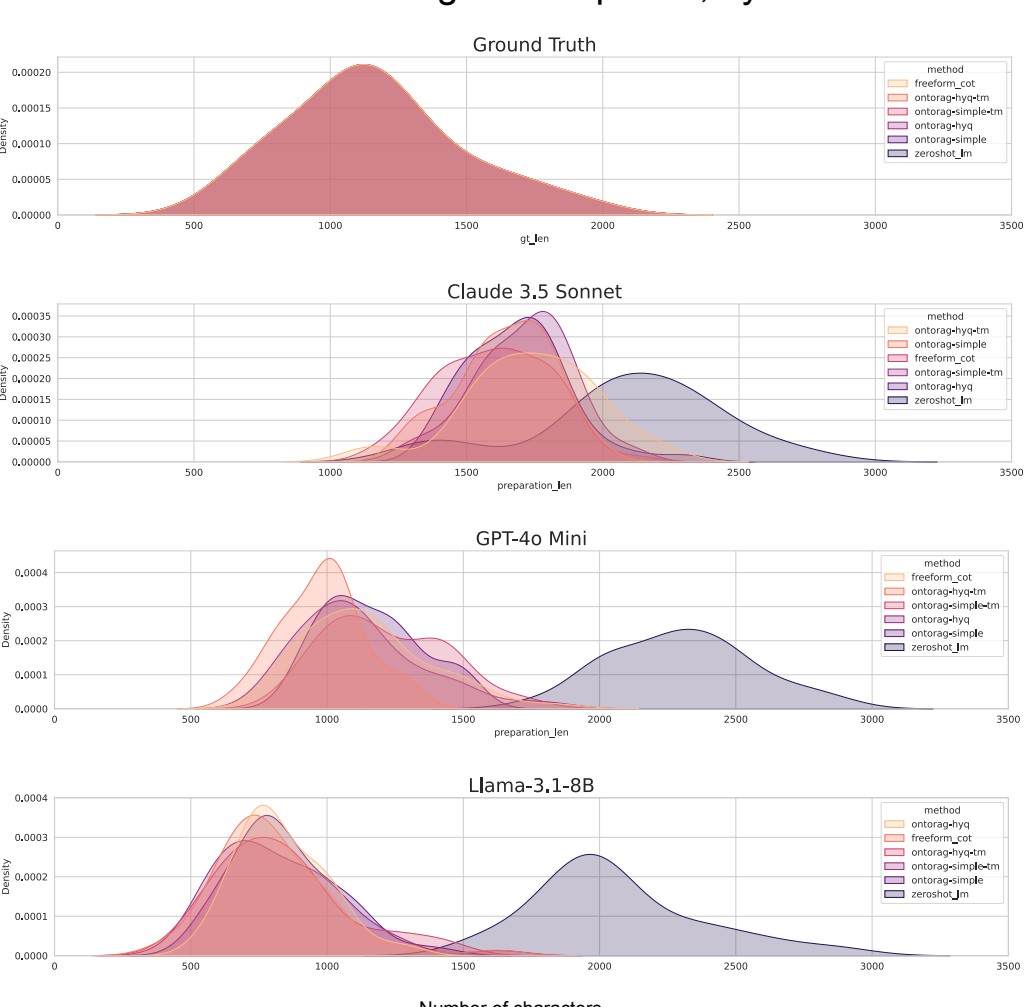

Figure 8: Distribution of response length for each LLM, by method. The plot shows a clear difference between the ZeroShot responses as compared to the rest of the methods.