# OpenReview forum: "Ontology-Retrieval Augmented Generation for Scientific Discovery"
_ICLR.cc/2025/Conference — Submitted to ICLR 2025_

### Official Review · Reviewer_4Tdh · 2024-10-17

**Soundness:** 2
**Presentation:** 3
**Contribution:** 2
**Rating:** 6
**Confidence:** 4

**Summary:**

In this study, the authors design and use an automatic ontology generator, OntoGen, to create ontologies for specialized domains in which there are no pre-existing ontologies. Then, they incorporate the generated ontology from OntoGen as input into OntoRAG, a retrieval-augmented system for LLMs. The authors aim to create a system which produces more accurate scientific output than LLMs or RAG systems. They first test OntoRAG without OntoGen, using biomedical ontologies in its place, for biomedical prediction tasks. Thereafter, they evaluate OntoRAG + OntoGen on a materials science application (single atom catalyst (SAC) synthesis) for which they designed a novel benchmark dataset. OntoRAG performs consistently better than a baseline RAG system for SAC synthesis. The novelty of this paper lies in (1) the design of an automatic ontology generator, OntoGen, (2) the development of a RAG system which incorporates ontologies as input, OntoRAG, and (3) the creation of a benchmark dataset, SACBench, for assessing LLM output within the context of a specialized, materials science domain.

**Strengths:**

- Novelty: The authors addressed a novel, interdisciplinary area between AI and the natural sciences.

- Clarity: The authors have done a great job of explaining the necessary background in a concise way. I commend the authors for acknowledging the significance and challenges which go along with applying AI approaches to scientific domains, in which plausibility is not necessarily the same as scientific accuracy.

- Creativity: The ideas in this paper are creative: the authors have found a unique way to address ongoing concerns surrounding LLM hallucinations, particularly within a scientific context.

- Reproducibility: The paper is generally well-written and easy to read. For the most part, the paper was clear, and experiments seem reproducible based on the details given.

**Weaknesses:**

I have two major concerns regarding gaps in the evaluation processes. These concerns make it difficult to confirm the scientific rigor of this study:

1. The authors should evaluate the capabilities of OntoGen against existing scientific ontologies, like the Gene Ontology. This will allow the reader to assess whether OntoGen can really produce ontologies that capture scientifically significant patterns. The authors could accomplish this through a comparison of the metrics reported in Fig. 3 or through metrics such as concept coverage, structural similarity, or expert evaluation of key relationships.

2. The experimental results in Figure 4 should also include the base LLMs, without any augmentation, as baselines. Specifically, the authors should report the performance metrics of the base LLMs on SACBench using the same criteria as OntoRAG. This will allow the reader to assess whether OntoRAG truly improves upon LLM accuracy within specialized, scientific domains.

**Expansions:**

1 (expansion): I do not think the authors sufficiently evaluated the capabilities of OntoGen before moving on to evaluate OntoRAG. Since OntoRAG on the SACBench dataset relies upon the output of OntoGen, it is necessary to ensure that OntoGen can produce ontologies with qualities consistent to established ones. Specifically, the authors should compare the output of OntoGen to an existing ontology. While there is no existing ontology for SAC, the authors acknowledge in Section 2.1 that other curated ontologies exist for other domains, like genetic or biomedical ones. For example, the authors could use OntoGen on a corpus of genetic literature and compare the generated ontology to the Gene Ontology.

2 (expansion): The experimental results given in Figure 4 are missing a key baseline: the base LLMs without any RAG system. The authors should include this baseline as it is critical to assess one of the aims of the paper ("enhancing the scientific accuracy of LLM outputs"). Additionally, this baseline is particularly important in light of the results of Section 5, in which the OntoRAG system performed worse than the base LLMs in a majority (6/10) of cases (based on the metrics reported Appendix A.0.1). The results of Section 5 (Appendix A.0.1) call to question why the authors decided to move on with OntoRAG + OntoGen. It appears that OntoRAG with pre-existing ontologies has no improvement or limited improvement over the base LLMs. If OntoRAG with established ontologies offers no substantial improvement, then the authors should clarify why they believe that OntoRAG + OntoGen will offer improvements. Specifically, the authors may be able to justify the use of OntoRAG + OntoGen by including the performances of base LLMs on SACBench.

**Questions:**

1. Have the authors conducted any further investigations into the results of Section 5? Specifically, the authors speculate that the results are due to discrepancies in vocabulary; have they confirmed that?

2. What is the verification process "to ensure fidelity to the source text" described in Section 4.1? Can the authors specifically describe the details of (or cite, if using another approach) this verification process?

3. Furthermore, how are discrepancies handled when the above verification process encounters them? Can the authors give a specific example of how a discrepancy might be handled?

4. What self-consistency techniques were used in Section 4.2? Can the authors give specific details or citations for these techniques?

---

> ### Author Response · Authors · 2024-11-25
>
> We sincerely thank the reviewer for their thorough comments. We have revised our manuscript and addressed your questions in the following way:
>
> # Q1: the authors speculate that the results are due to discrepancies in vocabulary; have they confirmed that?
>
> We appreciate the reviewer’s comment and thank them for pointing out this weakness. To address the reviewer's concern, we chose to conduct further analysis to support our speculation on the effect of vocabulary discrepancies on the performance of the system. We conducted a simple analysis on the correlation between the ontological relevance of the statement (i.e. how many ontological concepts are detected in the statement) and the performance of OntoRAG on evaluating said statements. The results are given in the following table.
>
> | Benchmark | Correlation |
> |-----------|-------------|
> | medqa     | 0.7852     |
> | mmlumed   | 0.7506      |
> | medmcqa   | 0.1018      |
>
> The results indicate an overall positive and strong correlation between ontological relevance and downstream performance. We again thank the reviewer for their question, and hope this addresses this point.
>
>
> # Q2: What is the verification process "to ensure fidelity to the source text" described in Section 4.1?
>
> thank you for this, indeed we have not made it very clear. We have updated the manuscript to make it clear that this process is a string matching operation, to ensure all the concepts proposed by the LLM are indeed sourced from the papers given as context, making it possible to detect and remove hallucinations.
>
> # Q3: how are discrepancies handled when the above verification process encounters them? Can the authors give a specific example of how a discrepancy might be handled?
>
> Since the verification process involves a string matching over the original text, discrepancies are handled by simply discarding the term from the vocabulary if it is not found. For instance, if the original text contains the term "carbon dioxide" but the LLM hallucinates "CO2", the latter term will be discarded from the vocabulary, even if it is a valid synonym. This is done as a countermeasure to avoid the introduction of hallucinated terms into the ontology.
>
> # Q4: What self-consistency techniques were used in Section 4.2?
>
> We have included details in the Appendix, where we formally describe and cite self-consistency, and where we describe its application in our work:
>
> "self-consistency is applied in the category generation step of Ontogen by generating multiple lists of categories and then taking the most frequent categories (i.e. the majority vote) as the final list. In the taxonomy extraction step, self-consistency is applied in the "query_relationships" function (see Algorithm 1 in Appendix C). In this case, a query is prompted $N$ times to the LLM (e.g. "Single-Atom Catalist isA ?"), and a taxonomic relationship (e.g. "Single-Atom Catalist isA Catalist") is extracted only if it is the answer in the majority of the $N$ queries (i.e. if a relationship appears in at least $(N + 1)/2$ answers).
>
> # Extra
>
> Regarding some of the concerns you raised on the weaknesses sections, here are a few clarifications, that we would gladly incorporate in the manuscript if the reviewer finds it appropriate.
>
> For the question "Can OntoGen produce ontologies that capture scient. significant patterns?" we have included slices of the generated Ontologies for SACs to show what patterns are encoded there, and analyze the patterns found there. We believe this analysis, along with the results from downstream applications (e.g. the SACBench) will be enough to show how good the results from OntoGen are.
>
> We have also included the base LLM (under the name of ZeroShot) as one of out baselines in the updated manuscript. Thank you for pointing this out, it is indeed a clear miss in our original submission.
>
> Finally, the biomedical benchmarks were run initially kind of as control experiments. No improvement in these would just show that using OntoRAG doesn't hurt, while it can help in tasks more related to scientific discovery, as we show with the SACBench.
>
> Still, we have re-run the experiments on the medical benchmarks, this time also including the gene ontology, and we have updated the results as follows:
>
> | Method | medmcqa | medqa | mmlumed |
> |--------|------------------|----------------|------------------|
> | zeroshot | 62.06 | 67.16 | 80.06 |
> | cot | 60.91 | **69.99** | 76.70 |
> | ontorag-simple | **64.12** | 68.34 | 79.26 |
> | ontorag-tm | 61.80 | 68.11 | 80.01 |
> | ontorag-hypo_ans | **64.04** | 67.64 | 79.96 |
> | ontorag-hypo_ans-tm | 62.13 | **69.36** | **80.65** |
>
> As can be seen in the new results, there's typically either an advantage or simply no improvement over the baselines (ZeroShot or CoT). With this, we update our analysis to account for this fact and make it clear that this works more as a control experiment.
>
> I hope the reviewer finds these updates reasonable, and we're open to hear more feedback to improve our work.

---

> > ### Comment · Reviewer_4Tdh · 2024-11-28
> > **Response to Rebuttal**
> >
> > Dear, Authors,
> >
> > I will raise my score to a 6 because you've addressed my second major concern (weakness \#2). Thank you for that.
> >
> > I do not feel the first concern (weakness \#1), regarding the evaluation of OntoRAG, is sufficiently addressed. I would like to leave additional explanations and feedback in case the authors wish to further improve their work.
> >
> > The information presented in Fig. 3 is not enough to evaluate the quality of an ontology- especially one that is generated by an LLM and not an expert.
> >
> > I also see that Reviewers **2Kkq** and **ee63** express similar concerns to the first weakness I listed. Regarding what Reviewer **ee63** said...:
> >
> >         "Because the LLMs-generated ontologies cannot utilized to enhance the LLMs directly without the human’s curation, since the hallucinations still exist when generating the ontologies with LLMs, the manual validator is needed."
> >
> > As an alternative to this, I believe another suitable alternative to a human-in-the-loop, which Reviewer **ee63** had suggested, would be to compare an ontology generated by OntoGen against a gold-standard ontology like the GO, as I suggested. While this would not be a direct evaluation of the SAC ontology, it would be convincing evidence toward the idea that OntoGen can produce high-quality ontologies.
> >
> > Additionally, regarding the biomedical benchmarks: "using OntoRAG doesn't hurt" is not a very convincing reason to go forward and apply OntoRAG to a domain in which there is a greater level of uncertainty (i.e., lack of information and no gold-standard ontology). I truly believe that this section confuses and derails the overall story being conveyed in this paper.
> >
> > Overall, however, I think the contents of the study are interesting, and I thank the authors for their efforts during the rebuttal.

---

### Official Review · Reviewer_2Kkq · 2024-10-30

**Soundness:** 2
**Presentation:** 3
**Contribution:** 2
**Rating:** 5
**Confidence:** 4

**Summary:**

This paper introduces OntoRAG, a novel approach that enhances Retrieval Augmented Generation (RAG) by incorporating ontological knowledge to improve the accuracy and scientific grounding of large language models (LLMs). The authors also present OntoGen, a tool for automatic ontology generation to extend OntoRAG's utility to fields without pre-existing ontologies. The key contributions are:
    • OntoRAG: An extension of RAG that retrieves and integrates relevant ontological information to improve reasoning and reduce hallucinations in large language models (LLMs).
    • OntoGen: An LLM-based pipeline for automatically constructing domain-specific ontologies from scientific papers.
    • SACBench: A benchmark for evaluating the synthesis of single-atom catalysts (SACs), used to test the OntoRAG approach in an emerging scientific domain.
The authors evaluate OntoRAG on standard biomedical benchmarks and the novel of Single-Atom Catalysis SACBench. Results show improvements over baseline RAG in some domains, reduction of hallucinations in LLMs, particularly for the SAC synthesis task.

**Strengths:**

• Novel approach: Combining ontologies with RAG is an innovative idea to enhance LLM performance in specialized scientific domains.
• Automatic ontology generation: OntoGen addresses a key bottleneck by automating the creation of ontologies for emerging fields.
• Application in Emerging Domains: The case study in Single-Atom Catalysis demonstrates the potential of the approach to aid scientific progress in cutting-edge fields where ontologies are not yet fully established.
• Reduction of Hallucinations: OntoRAG addresses a critical problem in LLMs - factual inaccuracies - by grounding the outputs in established scientific relationships and concepts.

**Weaknesses:**

• Weak evaluation: The authors test their approach on established benchmarks without significant improvement and a novel task in an emerging field that shows promise. Nevertheless, its only one benchmark in a specific field that shows some results.
• Limited Improvement in Aggregate Performance: Despite the benefits of ontology integration, the paper notes that the aggregate improvement across benchmarks is modest, suggesting that the effectiveness of OntoRAG depends on the specific domain.
• Ontology quality assessment: The paper lacks a thorough evaluation of the quality of automatically generated ontologies beyond downstream task performance.
• Computational Overhead: The process of ontology generation and integration adds complexity and computational cost to the pipeline, which may limit its practical use in certain scenarios.
• Expert Dependency: While OntoGen attempts to automate ontology creation, the variability between LLMs and the need for manual curation still imply a dependence on human expertise for high-quality outputs.

**Questions:**

• Can authors add additional benchmarks that show improvement in performance?
• How does OntoRAG handle conflicting information from different retrieved sources within a single ontology or between multiple ontologies?
• How does the quality of automatically generated ontologies compare to expert-curated ones in established fields?
• How sensitive is the performance of OntoRAG to the specific choices made in the ontology retrieval and fusion steps?

---

> ### Author Response · Authors · 2024-11-26
>
> We sincerely thank the reviewer for their thorough and insightful comments. We have carefully considered your comments and have made appropriate changes as we describe below, along with answers with your questions and clarifications.
>
> # Questions:
>
> ## Can authors add additional benchmarks that show improvement in performance?
>
> Indeed, the original results we show were not a clear improvement over the baselines. From the time of submission of the original work, we have changed prompts and slightly optimized the pipeline. Our results for the same benchmarks are now as shown in the following results table:
>
> | Method | medmcqa | medqa | mmlumed |
> |--------|------------------|----------------|------------------|
> | zeroshot | 62.06 | 67.16 | 80.06 |
> | cot | 60.91 | 69.99 | 76.70 |
> | ontorag-simple | **64.12** | 68.34 | 79.26 |
> | ontorag-tm | 61.80 | 68.11 | 80.01 |
> | ontorag-hypo_ans | **64.04** | 67.64 | 79.96 |
> | ontorag-hypo_ans-tm | 62.13 | **69.36** | **80.65** |
>
> As shown in the new results, OntoRAG performs typically on par or better than the zero-shot LLM, or even a CoT baseline. However please note that these benchmark experiments were included more as control experiments, to show that the use of OntoRAG does not deteriorate performance. The main results however are those on SACBench, which we augment and expand also in the updated version.
>
>
> ## How does OntoRAG handle conflicting information from different retrieved sources within a single ontology or between multiple ontologies?
>
> This is indeed a very good question, and we think it would be worth considering in a follow-up study. Indeed, conflicting information can be retrieved from different ontologies. However we argue that retrieved information from an existing ontology should be self-consistent (in the sense of [1]), already preventing these situations from their creation.
> In the case of ontologies generated with OntoGen, these situations are prevented by using a series of verification and control steps, where responses are required to be self consistent, as well as matching with the retrieved literature. We have elaborated and explained more of this in the manuscript thanks to this and other reviewer's comments.
>
>
> ## How does the quality of automatically generated ontologies compare to expert-curated ones in established fields?
>
> This is a very good question. We believe it would indeed be very interesting to e.g. generate a new gene ontology from collected papers in the field, however we note that the extensive amount of literature that has been produced for this field, and thus a massive number of terms associated, makes it a much more challenging undertaking, as compared against the field of SAC, and thus a comparison against the Gene Ontology is rather unfeasible.
> The question highlights a very good point however, and as such we have included a new analysis section in the Appendix, where we display slices of the generated SAC ontologies (by different LLMs) and include assessments of their quality by chemists experts in the field.
> We hope this update will address this concern, and we're very much looking forward to your comments on this.
>
>
> ## Sensitivity OntoRAG to the specific choices made in the ontology retrieval and fusion steps?
>
> This is a great question, that we tried to directly address in form of ablations in our original submission, however the presentation was not very clear. With the updated results (Table above) we can also respond to this question by looking at what each variation of ontorag is.
> In particular, we ablate the fusion step using two variants of fusion, namely _simple_ and _tm_, which stands for "translation module".
> The method _simple_, consists of providing all the ontological context as a simple string in json format containing all information. The _tm_ is an intermediate module that summarizes the raw ontological context, and is made to distill it down to the relevant information for the query.
> Although in this aspect the results are not very conclusive, there seems to be a net positive effect of using _tm_ on these benchmarks. In fact, the most clear trend that we find here (and also in the SACBench results, that we are adding to the updated manuscript) is that OntoRAG+HyQ benefits the most from _tm_, while OntoRAG-simple works better without _tm_.
>
>
> ## Extra
>
> To your additional concern regarding increased overhead and computational cost, our method is indeed more costly than simply using ZeroShot inference. However it's important to note that the ontology is only generated once, and the goal of it is to condense a lot of the state of a field, including concepts and relationships. In that sense, the ontology can continue being useful for other tasks, without needing to carry this overhead for every use.
>
> We hope our responses have addressed your concerns, and we stay open to further discussion if needed and appreciate any further feedback.
>
> [1] ArXiv, abs/2203.11171

---

> > ### Comment · Reviewer_2Kkq · 2024-11-26
> >
> > I appreciate the authors' detailed responses to the concerns raised in my initial review. While the authors have made efforts to address the concerns some of the issues are still there:
> >
> > The new benchmark results, while showing some improvements over the initial submission, still demonstrate only modest gains over baselines. The improvements are incremental rather than substantial and in some cases perform similarly to baseline approaches. And this is still an assessment on the same benchmarks.
> >
> > The addition of expert assessment of generated SAC ontologies in the appendix is an improvement.
> >
> > The ablation studies clarify some aspects of the system's sensitivity to different components, but the results remain somewhat inconclusive regarding the benefits of different fusion approaches.
> >
> > While these updates strengthen certain aspects of the paper, the core concerns about limited evaluation breadth and modest performance improvements remain.

---

> > > ### Author Response · Authors · 2024-11-27
> > >
> > > Thank you very much for your kind and insightful response.
> > > We understand the points you raised here, and we would like to address them more concretely in the manuscript with your feedback.
> > > Just for clarification, the key idea we try to prove is that using ontologies can improve in tasks relevant for scientific discovery, hence we would like to make it very clear that the main results of our paper are those realted to SACBench.
> > > The biomed benchmarks were conducted more as control experiments, to show that using ontorag has no detrimental effect on other tasks. Indeed, our results show that ontorag improves or remains similarly performant as the baselines.
> > >
> > > We hope we can continue this productive discussion towards improving our paper, and we appreciate all your results so far.
> > > We hope the points that were addressed, as well as the updated manuscript, are considered in a new evaluation. Thank you very much again!

---

### Official Review · Reviewer_htBH · 2024-11-01

**Soundness:** 1
**Presentation:** 1
**Contribution:** 2
**Rating:** 3
**Confidence:** 4

**Summary:**

This paper introduces an ontology-based retrieval-augmented generation (RAG) pipeline designed to enhance scientific discovery by integrating ontology-based knowledge with language models. Additionally, the paper presents OntoGen, an automated ontology-generation method for fields where no ontology exists, extending OntoRAG's applicability to emerging domains. The proposed method is mainly evaluated on biomedical QA and catalyst synthesis benchmarks.

**Strengths:**

- This work is well-motivated by the need in many scientific domains for expert-curated knowledge that goes beyond document-level retrieval. It seeks to a new RAG pipeline by integrating ontologies, which are widely adopted knowledge bases for specific domains.

- The pipeline addresses cases where ontologies are unavailable, proposing an automated approach to ontology construction from documents.

**Weaknesses:**

- Definition 2.1 for ontology requires a signficant revision as it is unclear and contains inaccuracies:
  - An ontology can be described as a set of logical axioms that define relationships among entities (concepts, properties, and instance) in the ontology. This approach avoids separating axioms from relationships, as relationships should not be limited to triples alone.
  - It seems that the relationships set $\mathcal{R}$ refer to object properties, while the properties set $\mathcal{P}$ appears to denote data properties.
  - Additionally, the notation $\forall i \in \mathcal{I} \exists c \mid c \in \mathcal{C}$ needs clearer explanation. If the intent is to express that an instance $x$ belongs to a class $C$, it would be more accurate to write $C(x)$ or $x: C$.

- Definition of RAG in Equation (1) is inaccurate: As stated, this definition implies that each retrieved document influences the generation probability and is weighted by its relevance. However, in a standard (vanilla) RAG setting, this is not the case; only a subset of retrieved documents typically impacts the generation process, without automatic weighting by relevance.

- The OntoRAG definition needs a significant revision since it builds on the earlier definitions of ontology and RAG, which contain inaccuracies.

- The OntoRAG methodology section lacks sufficient detail for reproduction. To enhance clarity, it would be helpful to include step-by-step explanations of the methodology components and provide running examples.

- The main evaluation in Table 1 primarily examines variations of OntoRAG and one Chain-of-Thought (CoT) baseline. However, it overlooks important comparisons with existing GraphRAG approaches, which similarly aim to incorporate graphs and knowledge bases within the RAG framework.

**Questions:**

**Suggestion/Typo**:

- In Table 1, the word “typo” appears to be a typo itself and may need correction.
- I recommend a careful review of formal definitions throughout the paper. For established concepts like "ontology," it would be beneficial to reference widely accepted definitions, such as those based on description logic. For processes like RAG, ensure the level of abstraction aligns with practical implementations. The current definition assumes independent and automatic weighting of each retrieved document, which is not universally applicable in RAG and oversimplifies the underlying mechanics.

---

> ### Author Response · Authors · 2024-11-22
> **Revision of definitions and notations used throughout the paper.**
>
> We sincerely thank the reviewer for their thorough and insightful comments. We have carefully considered your comments and revised the definitions in our manuscript accordingly. Below we present the revised definitions, and we hope we can iterate on this and give them an ideal shape for our work.
>
>
> ### Ontology
>
> For the definition of ontology, we now avoid the distinction between axioms and relationships. This aligns more with the implementation of ontology used in our work. In addition we have improved the notation by using $i:C, C \in \mathcal{C}$ to denote that instance i belongs to class C. Finally, we specify what are object properties (when defining I) and data properties (when defining P).
> Please find the full updated definition here:
>
> ---
> ---
>
> An ontology is a tuple $ \\{ \\mathcal{C}, \\mathcal{R}, \\mathcal{I}, \\mathcal{P} \\} $  where:
>
> - $\\mathcal{C}$ is a set of classes $\\{ C_{1}, C_{2},...,C_{n} \\}$ present in the ontology.
> - $\\mathcal{R}$ is a set of relationships present in the ontology.
>
>   $\\mathcal{R} = \\{ (C_{i}, r_{s}, C_{j}) | r_{s} \in \\mathcal{R}_s \\}$, where $\\mathcal{R}_s$ is the set of all possible relations (object properties).
> - $\\mathcal{I}$ is the set of all instances of classes present in the ontology
>
>   $\\mathcal{I} = \\{ i_{1},i_{2},...,i_{m} \\}$; $i:C, C \in \\mathcal{C}$.
> - $\\mathcal{P}$ is the set of all possible properties in an ontology.
>
>   $\\mathcal{P} = \\{  p_{1}, p_{2},...,p_{l}  \\} $ and $p:\\mathcal{I} \\xrightarrow{}\\mathcal{V}$ or $p:\\mathcal{C} \\xrightarrow{}\\mathcal{V}$; where $\\mathcal{V}$ is the set of all possible values for a property (data properties).
>
> ---
> ---
>
> Regarding our definition of RAG, while we took it from reference [1], it is indeed inaccurate for our implementation and we have modified it as follows to correct for this. It is now formulated in terms of retrieval function $R$ and fusion operator $F$. This way we directly address the fact that we only take a limited number (k) of retrieved documents, and perform no weighting on them. The probability of generation of a response $y$ is then directly affected by a single prior $F(x, R(x) )$.
> Please find our updated definition below:
>
> ---
> ---
>
>
> $p(y|x) = p_\\theta (y|F(x, R(x)))$. (1)
>
> with
>
> $R(x) = \arg \max_{z\in Z} ^k  \{r(z, x)\}$. (2)
>
> Where:
>
> - $p(y|x)$ is the probability of generating output y given input x.
> - $R(x)$ hence defines a set of the $k$ most relevant documents to $x$ under relevance function $r$.
> - $r$ is a _document relevance_ function, such that $r(z, x)$ quantifies the relevance of document $z$ to query $x$.
> - $F$ is a fusion operator.
> - $p_\theta (y|w)$  is the probability of generating $y$ given context $w$ for a language model parameterized by $\theta$.
>
> ---
> ---
>
> The definition for OntoRAG thus changes as follows:
>
> ---
> ---
>
> $p(y|x) = p_\theta (y|F(x, R(x), R_O(x)) )$. (3)
>
> with
>
> $R_O(x) = \{ O(c):  c \in C(x) \}$. (4)
>
> Where Eq. 1 is modified in Eq. 2 to include:
>
>
> - $R_O(x)$ is the ontological context relevant to query $x$, which depends on:
> - $O(c)$ is some ontological context retriever, and
> - $C(x)$ is a set of concepts found in text $x$.
>
> ---
> ---
>
> We will address the other points you have raised in another response. Thank you very much again for your feedback, we would greatly appreciate your thoughts on these revised definitions, to improve our paper.
>
>
>
> ### References
> [1] Lewis, P., Perez, E., Piktus, A., Petroni, F., Karpukhin, V., Goyal, N., Kuttler, H., Lewis, M., Yih, W., Rocktäschel, T., Riedel, S., & Kiela, D. (2020). Retrieval-Augmented Generation for Knowledge-Intensive NLP Tasks. ArXiv, abs/2005.11401.

---

> > ### Comment · Reviewer_htBH · 2024-11-23
> >
> > I appreciate the authors’ effort to revise and correct their definitions. While these definitions may not fully align with standard conventions and can be further polished, they are now at least reasonable.
> >
> > From my understanding in [1], this does not represent a naive retrieve-and-paste RAG setting. While it is valid to define RAG within the context of your specific approach, I do not believe that a general probabilistic definition for RAG is feasible.
> >
> > That said, aside from revising the definitions, I have not seen the authors address the other concerns raised in my initial review.

---

> > > ### Author Response · Authors · 2024-11-24
> > >
> > > Thank you very much for your comments. We're happy the definitions are more acceptable now. If there's any other specific edit you think is pertinent, we would be happy to further review our definitions.
> > >
> > > # Definition of RAG and OntoRAG.
> > > For the definition of RAG, an thus OntoRAG, we have opted for a probabilistic definition (following [1]) as a given answer is never deterministically generated, but is instead subject to a decoding process. However we think the definition could be further adapted to include a hint to the sampling process, something in the lines of:
> > >
> > > ---
> > > $y \sim p(y|x) = p_\theta(y| F(x, R(x))$
> > > ---
> > >
> > > This definition of course doesn't aim to generally describe RAG, but we believe it fits the needs of our manuscript.
> > > We are open to any further suggestions and discussion.
> > >
> > >
> > > Regarding the other comments/questions:
> > >
> > > ---
> > >
> > > # Details for reproduction
> > >
> > > To improve the clarity and reproducibility of our work, we are updating the paper to include pseudo-code for the proposed ontorag methodology, as well as step-by-step explanations with running examples in the appendix. This way we ensure the readers understand what the ontological context is, how it's being used in the pipeline, and how it can be helpful for performance as we show.
> > > In addition, we are releasing the code associated with or paper.
> > > - ontorag
> > > - the code for running the sac-specific experiments (in another repo)
> > >
> > > Here is the code:
> > > https://figshare.com/s/4f898ef092ae5898c1b7
> > >
> > > And we have added additional pseudo-code snippets to the Appendix, illustrating:
> > > - the process of retrieval of ontological context
> > > - a description of the flow of information from query, to retrieval, fusion, and finally, response generation.
> > >
> > > We will update this in the manuscript.
> > >
> > >
> > > ---
> > >
> > > # Comparisons with other baselines
> > >
> > > We understand the reviewer's concern regarding comparisons with existing GraphRAG approaches.
> > > For a more complete comparison against other baselines, we have now included a comparison against the "raw" LLM (ZeroShot), which is a common baseline for this type of studies.
> > > We understand the reviewer's concern, however we need to note that GraphRAG is a method that leverages Knowledge Graphs (KGs) for RAG with LLMs. While this is similar in that both approaches aim to incorporate knowledge bases in the RAG framework, our approach relies on ontologies, which are representations of conceptualizations of knowledge, rather than graphs of semantic triples in the form of KGs. We thus believe that including such a comparison is out of the scope of this work.
> > >
> > >
> > > We hope these explainations clarify our approach, ideas, and experimental design, and we remain open for any further question or feedback you might have!

---

> > > > ### Comment · Reviewer_htBH · 2024-11-24
> > > >
> > > > I respectfully disagree with the authors’ opinion that GraphRAG is irrelevant. While it is true that GraphRAG typically targets knowledge graphs (KGs) and ontologies often capture richer semantics than KGs, ontologies can still be represented in a KG format through standards such as RDF and RDFS. Numerous ontology projection methods are available to facilitate this conversion. **Even in your definition**, you use triples to define relationships in ontologies, they are essentially forming KGs. From a broader perspective, the "Graph" in "GraphRAG" refers to structured data that can potentially be represented as graphs, rather than being limited to KGs alone.

---

> > > > > ### Author Response · Authors · 2024-11-29
> > > > >
> > > > > We again thank you sincerely for your comments and your effort in bringing our paper to a better shape.
> > > > > We understand the concern with GraphRAG, and also we would like to highlight that the rest of the comments were addressed. Please refer to the updated manuscript to see these and many other improvements we have implemented based on yours and other author's comments.
> > > > > We would like to invite you to consider this in submitting an updated evaluation. We will continue strengthening our work and again, we appreciate your comments and feedback.

---

### Official Review · Reviewer_ee63 · 2024-11-04

**Soundness:** 3
**Presentation:** 3
**Contribution:** 2
**Rating:** 5
**Confidence:** 4

**Summary:**

This paper presents an OntoRAG that leverages the LLMs generated ontologies as the context to enhance the RAG by retrieving taxonomical knowledge from context for accelerating scientific discovery.  The results on the SACBench benchmark demonstrate that OntoRAG outperforms the CoT-based RAG on accuracy, completeness, and order. Additionally, the quality of the ontologies generated by LLMs is evaluated by the downstream task on the biomedical QA benchmark.

The paper is well-written and organized, but the pipeline of ontology generation (OntoGen) and RAG with ontologies (OntoRAG) is not a novel contribution, as the existing methods have already investigated it.

My main concern is that the ontologies generated by LLMs cannot be utilized to enhance the LLMs directly without the human’s curation since hallucinations remain when generating the ontologies with LLMs.

**Strengths:**

S1. The paper is well-written and organized, and the methodology of OntoRAG is well-designed and demonstrated.

S2. The OntoGen pipeline is created to generate the ontologies based on multiple calling the long-context LLMs.

S3. The experiments on the SACBench benchmark and biomedical QA benchmark are conducted to evaluate the performance of OntoRAG and the quality of LLM-generated ontologies.

**Weaknesses:**

W1. The presented OntoGen and OntoRAG pipeline is not novel, as the existing works have already been investigated but they are missed in related works. (Details in Q1)

W2. The ontologies generated by LLMs are utilized directly as context for RAG without the human’s validations. (Details in Q2)

W3. The source code and claimed SACBench benchmark dataset are not provided for reproducibility.

W4. The readability of this paper needs to be improved, as some results analyses and the ablation study are missed. (Details in Q3)

W5. Some typos need to be fixed and avoided. For example, the parentheses after “axioms” should be removed “axioms()-> axioms” in line 289-290, the comma after “in” should be removed “in. order to-> in order to”  in line 408, the “ACcuracy->Accuracy” in line 916, etc.

**Questions:**

Q1: How differ of your proposed OntoRAG  and OntoGen pipeline  when comparing with the existing DRAGON-AI (https://arxiv.org/abs/2312.10904) and LLMs4OL (https://link.springer.com/chapter/10.1007/978-3-031-47240-4_22)?

Q2: Can you provide the details of your verification process and manual effort for LLM-generated ontologies that you mentioned in Lines 340-341 and 363-364?
Because the LLMs-generated ontologies cannot utilized to enhance the LLMs directly without the human’s curation, since the hallucinations still exist when generating the ontologies with LLMs, the manual validator is needed.

Q3: Can you provide a detailed analysis of the results that are reported in Table 2 and Table 3 and highlight how much the OntoGen and OntoRAG contribute to the final results?

---

> ### Author Response · Authors · 2024-11-25
>
> We sincerely thank the reviewer for their thorough and insightful comments. We have carefully considered your comments and revised the definitions in our manuscript. Below we address the questions you have posed, hoping to improve the quality and clarity of our work:
>
> # Q1: OntoRAG and OntoGen vs existing DRAGON-AI and LLMs4OL?
>
> Thank you very much for highlighting these works, indeed we have missed to reference them, and we would like to clarify the differences between our work and these others.
>
> On the one hand, LLMs4OL focuses on the evaluation of LLMs rather than the generation itself of ontologies. In this work an LLM is prompted to assess whether a term is a subclass of another, without additional context. DRAGON-AI on the other hand tackles the task of inserting new terms into an existing ontology, so it requires an existing ontology. Our work, particularly OntoGen, takes a set of documents as an input, and generates an ontology by selecting and interconnecting terms into an ontology.
>
> We address this in our updated manuscript by directly mentioning what they do and how our work is different, in the Introduction section. In particular, we add the following lines:
>
> "Tools have been proposed recently to accelerate this process by inserting new terms into an already existing ontology Toro et al. (2023); Funk et al. (2023), or automating tasks such as term typing, taxonomy discovery, etc, under the frame of Ontology Learning Ciatto et al. (2024); Toro et al. (2023); Babaei Giglou et al. (2023). However, none of these works has attempted to generate full ontologies. To address this we propose OntoGen, an LLM-based method for automatic end-to-end generation of ontologies."
>
> We hope this helps clarify the novelty issue you have raised.
>
>
> # Q2: Details of verification process and manual effort for LLM-generated ontologies?
>
> Thank you very much for your comment, indeed some of these details have been left unspecified.
> For both the verification process and the comment on manual effort, we have added new sections to the appendix that elaborates further on the role and details of these processes.
> To clarify here, the verification process consists of a string match that checks whether each of the terms in the current list, indeed exists in the documents that were provided to the LLM as context.
>
> Regarding the comment on manual effort, we clarify that this is only made with the "seed" terms used to initialize the taxonomy. This seed list has only few terms as this is automatically extracted and is composed of only the most common terms extracted by the LLM; in the case of the SAC ontology this was around 7 terms, namely _Characterization, Physical properties, Synthesis methods, Reaction mechanisms, Structure, Applications, Reactions_ and _Support_. The "manual curation" we perform in this step involved selecting the following additional categories from the pool of generated categories, so as to make the ontology more aligned with our chemistry knowledge: _Catalytic performance, Preparation methods, Theory and modelling_ , and _Materials_.
>
> This is, the manual effort that is expected here to exclude one or more categories from the generated list or include additional categories if needed. Notice that this does not involve manually refining the whole taxonomy, but just the set of terms from the initial seed.
>
>
> We hope this can help clarify any concerns regarding human involvement. Indeed, no extensive human curation is required at any point, only shortly at the early stages for more complete results. Regarding hallucinations, countermeasures have been taken through the generation process so as to minimize their impact. Just to recap, a verification step is performed after vocabulary extraction, while self-consistency is applied both during category generation and taxonomy extraction.
>
> # Q3: Analysis of the results that are reported in Table 2 and Table 3 and highlight how much the OntoGen and OntoRAG contribute to the final results?
>
> As the reviewer noted, our manuscript falls short to analyze these two tables and the effect of these 2 components in the final results. Thank you for pointing this out. In our updated manuscript we conducted additional experiments to assess the effect of variations of OntoRAG (Table 2). In particular, we also add the final results on these benchmarks to assess the effect of the fusion operator (F) with two variations: concat, and translation-module. We append the table in a follow-up comment.
>
> For Table 3, we again conducted additional experiments on a fixed version of OntoRAG, but using different ontologies (generated with different LLMs in OntoGen), to assess the effect of this part. In addition we have included more of the SACBench metrics for the sake of completeness, and we further analyze this in the manuscript.
>
>
> We thank again the reviewer for their insightful comments, and look forward to productive discussion.

---

### Author Response · Authors · 2024-11-29
**Final submission comments**

Dear Program Chairs, Area Chairs, and Reviewers,

We sincerely thank you for your thorough and insightful feedback. We have made substantial revisions to address your concerns and significantly improve our manuscript, submitted on this rebuttal:

- **Definitions and Notations:** We have revised our definitions of ontology, RAG, and OntoRAG to improve clarity and accuracy (thank you reviewer **htBH** for the feedback on this). See Definition 1 and Equations 1, 2 and 3.

- **Improved inclusion of baselines:** As suggested by the reviewers, we have included ZeroShot LLMs as a new baseline, which we overlooked in the original submission.

- **Biomedical benchmarks:** We have updated the evaluation results on the 3 biomedical benchmarks used in this paper. In the new manuscript, we show how OntoRAG methods perform on par or better than baselines (ZeroShot and CoT) in average (Table 1, Appendix A.1.1), and excell at tasks for which the provided ontologies are more relevant (in our experiments: genetics, anatomy, microbiology, see Table 2, Appendix A.2).

- **Additional analysis:** Furthermore, we perform an additional analysis where we show that ontological relevance (as measured by the average number of concepts retrieved by the retrieval operator) correlate strongly with improved performance on the biomedical benchmarks. See Appendix A.1.2, Table 2.

- **Quality assessment of Ontologies:** Some reviewers raised concerns regarding the reliability of the ontologies generated by OntoGen. We have added a new section in the Appendix (A.4.6) where we show some parts of the produced SAC ontologies with Llama-3.1-70B and Claude-3.5-Sonnet, along with an analysis by a domain expert (see Appendix A.4.5).

- **Ablation on ontology source:** To further this analysis, and with the argument that the best way to evaluate the quality of an ontology is through a downstream application, we report the results of running multiple methods (ontorag and baselines) using SAC ontologies generated by two different models (Llama-3.1-70B and Claude-3.5-Sonnet), see Appendix A.7, Tables 4 to 7. These results show that, overall the largest effect is on the "metal" and "support" metrics, with between 6%-10% points of difference, with the Claude-generated ontology achieving a higher score.

- **Methodology:** We have greatly upgraded our work by adding pseudo-code (Algorithm 1, 2, 3, 4 in the Appendix), along with code-snippets (Figure 5, Appendix) to improve the clarity and reproducbility of our work.

- **Code release:** We have made our code publicly available for further transparency and reproducibility. It is under https://figshare.com/s/4f898ef092ae5898c1b7 and we have updated our abstract accordingly.

- **Ablation studies:** We have improved and clarified our ablation studies on the fusion operators. Table 1, A.1.1 shows the downstream effect of using the translation module (TM) on OntoRAG, as evaluated by the biomedical benchmarks.

- **Further clarifications:** We have elaborated and clarified in the manuscript the details of the verification process of OntoGen, and the self-consistency techniques used there. See Appendix A.4.1, A.4.2.

- **SACBench clarifications:** We have further improved explainations for the metrics, and added more details on the generation and curation process (Appendix A.5). This is one of our key contributions, and we have released the code as part of the submission.

- **SACBench results:** Additionally, we add more results and analyses on SACBench, that help us understand how and where OntoRAG works better than baselines. In particular, see Figure 7 and 8 of the appendix.


These extensive revisions address most of the concerns raised by the reviewers, and significantly strengthen our work. We believe these improvements, along with the novelty of our contributions and their potential impact on scientific discovery, make a compelling case for acceptance. We look forward to your updated evaluation and remain open to any further feedback to ensure our paper meets ICLR's high standards.

---

### Meta-Review · Area_Chair_PNqS · 2024-12-22

**Metareview:**

This paper introduces an ontology-based retrieval-augmented generation (RAG) pipeline for integrating ontology-based knowledge with LLMs. Additionally, the paper presents OntoGen, an automated ontology-generation method for fields where ontologies don't exist.

The reviewers recognized several strengths, including:

- Problem being well motivated: Using ontology-deriven RAG is a nice solution to address LLMs inaccuracies in scientific discovery
- Novelty: OntoRAG innovatively integrates ontologies into the RAG framework to ground outputs in established relationships.
- Benchmark  SACBench could be a valuable contribution

However, the reviewers identified several major weaknesses in the paper, particularly in presentation, methodology, evaluation, and clarity. Reviewer ee63 noted missing results analyses and ablation studies, while htBH flagged inaccuracies in key definitions and insufficient detail for reproducing the approach. The risk of hallucinations in LLM-generated ontologies was a shared concern (ee63 and 2Kkq). And 4Tdh suggested validation against existing ontologies.

The evaluation was also criticized as limited and unconvincing. htBH and 4Tdh highlighted the absence of key baselines, including comparisons with graph-based methods and non-RAG LLMs. 2Kkq and 4Tdh questioned the empirical results and lack of evaluation breadth.

The authors tried to address some of these points during the rebuttal, but the reviewers aren't entirely convinced.

**Additional Comments On Reviewer Discussion:**

- Reviewer 4Tdh raised concerns about the lack of baseline evaluations and the insufficient validation of generated ontologies. The authors addressed the first concern but did not sufficiently resolve the latter.
- Reviewer ee63 emphasized the risk of hallucinations in generated ontologies and called for human validation. The authors clarified their verification process but did not offer a comprehensive resolution.
- Reviewer htBH criticized the paper's definitions and lack of comparisons with GraphRAG. While the authors revised definitions and added explanations, they argued that GraphRAG comparisons were out of scope.
- Reviewer 2Kkq noted modest improvements in performance and questioned the generalizability of OntoRAG. Despite updates, concerns about limited evaluation breadth persisted.

Overall, the reviewers acknowledged the authors’ efforts to address concerns but remained skeptical about the scientific rigor of the approach.

---

### Decision · Program_Chairs · 2025-01-22

Reject